EMBO
Molecular Medicine

# FLT1 activation in cancer cells promotes PARP-inhibitor resistance in breast cancer

Yifan Tai[1,8], Angela Chow[1,12], Seoyoung Han[1,9,12], Courtney Coker [1,12], Wanchao Ma[1,12], Yifan Gu[1], Valeria Estrada Navarro [2], Manoj Kandpal [3], Hanina Hibshoosh[4], Kevin Kalinsky[5], Katia Manova-Todorova [6,10], Anton Safonov[6], Elaine M Walsh[6,11], Mark Robson[6], Larry Norton[6], Richard Baer[1,4], Taha Merghoub[2], Anup K Biswas [1,4✉] & Swarnali Acharyya [1,4,7✉]

## Abstract

**Acquired resistance to PARP inhibitors (PARPi) remains a treatment challenge for *BRCA1/2*-mutant breast cancer that drastically shortens patient survival. Although several resistance mechanisms have been identified, none have been successfully targeted in the clinic. Using new PARPi-resistance models of *Brca1*- and *Bard1*-mutant breast cancer generated in-vivo, we identified FLT1 (VEGFR1) as a driver of resistance. Unlike the known role of VEGF signaling in angiogenesis, we demonstrate a novel, non-canonical role for FLT1 signaling that protects cancer cells from PARPi in-vivo through a combination of cell-intrinsic and cell-extrinsic pathways. We demonstrate that FLT1 blockade suppresses AKT activation, increases tumor infiltration of CD8+ T cells, and causes dramatic regression of PARPi-resistant breast tumors in a T-cell-dependent manner. Moreover, PARPi-resistant tumor cells can be readily re-sensitized to PARPi by targeting *Flt1* either genetically (*Flt1*-suppression) or pharmacologically (axitinib). Importantly, a retrospective series of breast cancer patients treated with PARPi demonstrated shorter progression-free survival in cases with FLT1 activation at pre-treatment. Our study therefore identifies FLT1 as a potential therapeutic target in PARPi-resistant, *BRCA1/2*-mutant breast cancer.**

**Keywords** FLT1; VEGFR1; Breast Cancer; PARP-Inhibitor-Resistance
**Subject Categories** Cancer; DNA Replication, Recombination & Repair

## Introduction

The tumor suppressor genes breast cancer 1 (*BRCA1*) and breast cancer 2 (*BRCA2*) maintain cellular genome integrity through various processes, including homologous recombination (HR)-mediated repair of DNA breaks (Groelly et al, 2023; Scully and Livingston, 2000; Venkitaraman, 2019; Xu et al, 1999). Individuals harboring heterozygous germline mutations in either *BRCA1* or *BRCA2* (collectively denoted as "*BRCA1/2*" hereafter) display heightened susceptibility to certain malignancies, especially breast and ovarian cancers (Ford et al, 1994; Ford et al, 1998) that arise upon loss of the remaining wild-type *BRCA1/2* alleles and the onset of extensive genome instability (Brose et al, 2002; Hall et al, 1990; Miki et al, 1994; van der Kolk et al, 2010). In addition to the germline mutations implicated in familial breast cancer, somatic *BRCA1/2* mutations are also detected in some sporadic cases of breast cancer (Nik-Zainal et al, 2016; Vidula et al, 2020). Most *BRCA1*-mutated breast tumors, and a subset of *BRCA2*-mutated tumors, present as triple-negative breast cancer (TNBC), which is associated with a poor prognosis and high likelihood of recurrence (Comen et al, 2011; Foulkes et al, 2004).

In cells that experience genotoxic stress, the poly(ADP-ribose) polymerases PARP1 and PARP2 promote the DNA damage response (DDR) by recognizing DNA breaks and by PARylating a variety of DDR factors, including those involved in single-strand DNA repair (Comen and Robson, 2010; D'Andrea, 2018; Dias et al, 2021). Since *BRCA1/2*-mutant tumor cells are deficient in HR-mediated DNA repair, they are especially reliant on PARP1/2 for their survival (Bryant et al, 2005; Farmer et al, 2005; Venkitaraman, 2019). The discovery of synthetic lethality between PARP enzymatic inhibition and *BRCA1/2* mutations ultimately led to the rapid clinical translation of PARP inhibitors (PARPi) for the treatment of *BRCA1/2*-mutant breast cancers (Bryant et al, 2005; Farmer et al, 2005). Among patients with *BRCA1/2*-mutant breast

[1]Institute for Cancer Genetics, 1130 St Nicholas Avenue, Columbia University Irving Medical Center, New York, NY 10032, USA. [2]Sandra and Edward Meyer Cancer Center, Weill Cornell Medicine, 1300 York Avenue, New York, NY 10065, USA. [3]Centre for Clinical and Translational Science, Rockefeller University Hospital, 1230 York Ave, New York, NY 10065, USA. [4]Department of Pathology and Cell Biology, 630 W 168th St, Columbia University Irving Medical Center, New York, NY 10032, USA. [5]Winship Cancer Institute of Emory University, Emory University School of Medicine, 1365 Clifton Road NE, Atlanta, GA 30322, USA. [6]Memorial Sloan-Kettering Cancer Center, 1275 York Avenue, New York, NY 10065, USA. [7]Herbert Irving Comprehensive Cancer Center, 1130 St. Nicholas Ave, Columbia University Irving Medical Center, New York, NY 10032, USA. [8]Present address: Department of Biology, McGill University, Montreal, Quebec, QC H3G0B1, Canada. [9]Present address: Jacobs School of Medicine, University of Buffalo, New York, NY, USA. [10]Present address: Drukier Institute for Children's Health, Meyer Cancer Center, Weill Cornell Medicine, New York, NY, USA. [11]Present address: Department of Medicine, Georgetown Lombardi Comprehensive Cancer Center, 3800 Reservoir Rd, NW, Washington DC 20007, USA. [12]These authors contributed equally: Angela Chow, Seoyoung Han, Courtney Coker, Wanchao Ma. ✉E-mail: akb2180@cumc.columbia.edu; sa3141@cumc.columbia.edu

cancer, dramatic initial responses to PARPi drugs, such as olaparib and talazoparib, led to their FDA approval as monotherapy (Comen and Robson, 2010; D'Andrea, 2018; Litton et al, 2018; Robson et al, 2017). However, responses were short-lived and typically resulted in lethal recurrences, thus prompting the search for mechanisms of PARPi resistance.

Early work established that PARPi resistance can develop in a subset of breast cancer patients by reversion mutations in the *BRCA1/2* genes that restore HR function and thereby abrogate cellular dependence on PARP1/2 (Barber et al, 2013; Lin et al, 2019; Pettitt et al, 2020; Tobalina et al, 2021; Waks et al, 2020). In addition, systematic analyses of PARPi resistance mechanisms using *BRCA1/2*-mutant cells, both in culture and in mice, have identified alterations in a variety of other DDR genes that can potentially promote PARPi resistance by restoring HR activity (Berti et al, 2020; Bhin et al, 2023; Cruz et al, 2018; D'Andrea, 2018; Dias et al, 2021; Drost et al, 2016; Henneman et al, 2015; Hobbs et al, 2021; McCabe et al, 2006; Powell, 2016; Rottenberg et al, 2008; Wang et al, 2016). Indeed, some of the DDR genes implicated in these experimental studies (e.g., *TP53BP1* and *MRE11A*) have also been observed in PARPi-resistant tumors from patients with *BRCA1/2*-mutant breast cancer (Waks et al, 2020). Despite this progress, circumventing PARPi resistance is not currently feasible in the clinic.

To identify alternative mechanisms of PARPi resistance, we generated orthotopic allografts using tumor cells derived from genetically engineered mouse models (GEMMs) of *Brca1*- or *Bard1*-deficient breast cancer (Shakya et al, 2008). Most BRCA1 functions, including HR-mediated DNA repair, are executed by the BRCA1/BARD1 heterodimer, a nuclear complex formed by BRCA1 and BRCA1-associated RING domain 1 (BARD1) proteins (Lim et al, 2023; Wu et al, 1996). Genetic inactivation of either *Brca1* or *Bard1* in mammary epithelial cells leads to the development of triple-negative carcinomas that are indistinguishable in latency, frequency, cytogenetic features, and histopathology (Shakya et al, 2008). We chose to study PARPi resistance using *Brca1*- and *Bard1*-deficient allograft models for two main reasons. First, these models recapitulate the typical clinical course of PARPi therapy, which is characterized by a striking initial response to treatment, the subsequent development of resistance, and finally, cancer progression (D'Andrea, 2018; Dias et al, 2021; Tung and Garber, 2022). Second, many experimental studies of PARPi resistance mechanisms have been conducted using cell lines or subcutaneous tumors (xenografts) in immunocompromised mice. Since PARPi treatment is known to impact both innate and adaptive immunity (Ding et al, 2018; Galindo-Campos et al, 2019; Moreno-Lama et al, 2020; Shen et al, 2019; Wang et al, 2022), we sought to model the clinical setting and physiological context in which PARPi resistance develops using the new *Brca1*- and *Bard1*-deficient treatment models.

Analogous to breast cancer patients with *BRCA1* mutations, mice bearing *Brca1*- and *Bard1*-deficient tumors display remarkable initial responses to PARPi that are inevitably followed by disease progression. As expected, the tumors that progress are no longer sensitive to PARPi treatment in-vivo. Surprisingly, however, the PARPi-resistant tumor cells retain their sensitivity to PARPi in vitro, suggesting that adaptive mechanisms operating in the tumor microenvironment protect these PARPi-sensitive cells from PARP inhibition in-vivo. In contrast to resistance mechanisms that

restore the DDR functions of BRCA1, our study identified FLT1 (VEGFR1) signaling as a novel driver of in-vivo PARPi resistance that can be therapeutically targeted. We show that FLT1 is activated in the tumor cells of PARPi-resistant tumors from *Brca1*- and *Bard1*-deficient models as well as from breast cancer patients. Mechanistically, FLT1 signaling protects these cells from PARPi-induced death by activating AKT pro-survival pathways and by dampening the cytotoxic immune response. Moreover, patients with *BRCA1/2*- or *PALB2* (partner and localizer of BRCA2)-mutant breast tumors that display heightened FLT1 expression in their tumor cells at pre-treatment are at high risk for progression on PARPi. Importantly, we observed that blocking FLT1 signaling, either genetically or pharmacologically, re-sensitizes PARPi-resistant tumors to PARPi treatment. Thus, by circumventing the development of PARPi resistance in breast cancer patients, FLT1 blockade may serve to unleash the full clinical potential of PARPi therapy.

## Results

### Generation of in-vivo PARPi-response-and-progression models of *Brca1*- and *Bard1*-deficient breast cancer

To study mechanisms of PARPi resistance in-vivo, we isolated *Brca1*- or *Bard1*-deficient breast tumor cells (hereafter referred to as "Brca1-def" and "Bard1-def") derived from *Brca1*- or *Bard1*-conditionally deleted mice (Shakya et al, 2008) and injected these cells in the mammary glands of syngeneic B6/129F1 mice (Fig. 1). Tumor-bearing mice were randomized and treated five days/week with either vehicle (referred to as "Veh" in the figures) or the PARPi talazoparib (referred to as "Tal" in the figures), an FDA-approved PARPi in human breast cancer, at a dose of 0.3 mg/kg body weight/day starting at either 14 days (*Brca1*-def) or 7 days (*Bard1*-def) post tumor-cell injection (see schematic in Fig. 1A,D). Tumor size was measured weekly, and tumors were collected either when they reached the size limit or when mice developed a body-conditioning score (BCS) of 2 or less, following the guidelines for euthanasia. Although PARPi treatment suppressed tumor growth for several weeks, tumors began to progress and eventually became refractory to the treatment in all treated *Brca1*-def mice (Fig. 1B). Analogous to the *Brca1*-def model, *Bard1*-def tumors also responded initially to PARPi but soon progressed (Fig. 1E). While tumor growth in *Brca1*-def models was monitored for 13 weeks post tumor-cell injection, *Bard1*-def models could only be monitored for 5 weeks due to the onset of cachexia, as described in our previous studies (Shakri et al, 2020). Tumor cells that were are refractory to PARPi (abbreviated as "PARPi-Res" in the figure) from both PARPi-treated *Brca1*-def and *Bard1*-def tumors were then isolated, re-injected into the mammary glands of new B6/129F1 recipient mice (Fig. 1C,F) and treated as described above. Notably, in contrast to the tumors generated from parental PARPi-sensitive tumor cells (i.e., PARPi-sensitive, abbreviated as "PARPi-Sen" in Fig. 1B,E), tumors derived from PARPi-treated mice (PARPi-Res tumors) failed to respond in-vivo to PARPi starting from the onset of treatment (Fig. 1C,F), confirming the acquisition of drug resistance. Surprisingly, while the difference in response to talazoparib between the PARPi-sensitive and PARPi-resistant tumor lines was striking in-vivo (Fig. 1B compared to 1C and 1E

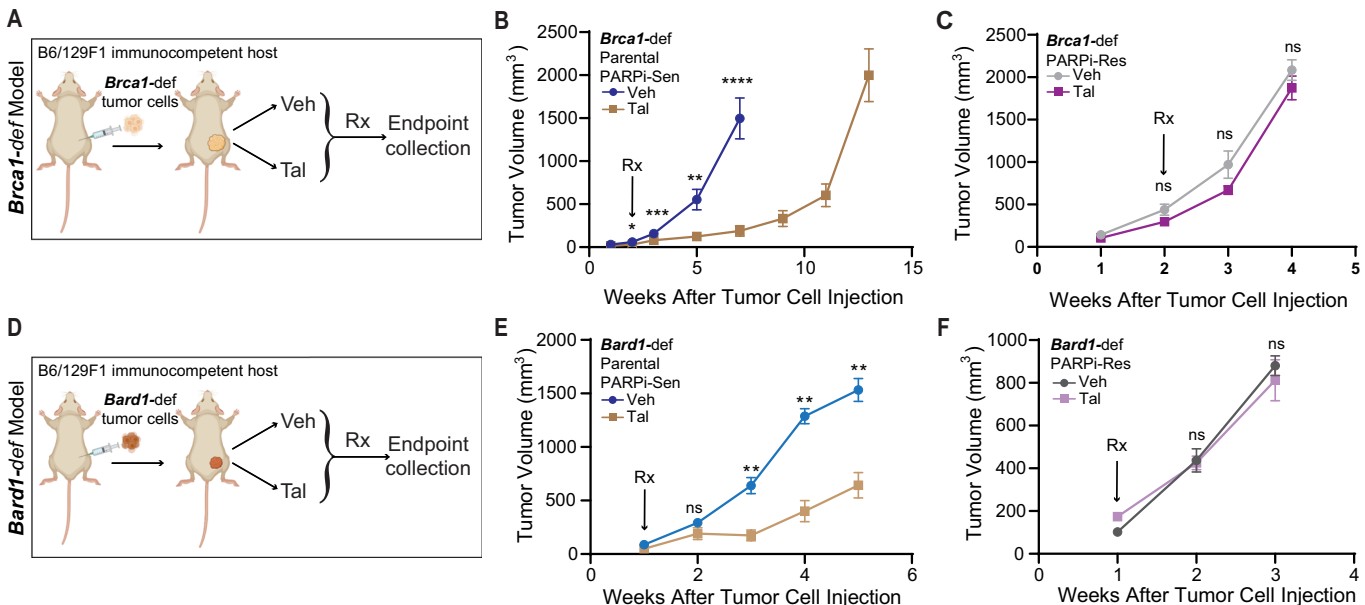

**Figure 1.   Generation of in-vivo talazoparib (PARPi)-response-and-progression *Brca1*-def and *Bard1*-def orthotopic models.**

(A) Schematic representation of the in-vivo treatment to generate talazoparib-resistant, *Brca1*-def breast tumor cells in mice. (B) Mice were injected with parental Tal-sensitive ("PARPi-Sen") *Brca1*-def breast tumor cells in the mammary gland of immunocompetent B6/129F1 mice, and tumor growth was monitored weekly. Mice were treated with either vehicle ("Veh") or talazoparib ("Tal"), beginning at 2 weeks after tumor-cell injection when tumors were palpable. *Brca1*-def parental tumors responded to PARPi until 7 weeks (drug-response phase), after which tumors became drug-refractory and re-grew (progression phase). Tal-refractory tumors were then harvested at 13 weeks after tumor-cell injection, and Tal-resistant ("PARPi-Res") tumor cells were isolated. $n = 24$ for Veh-treated mice; $n = 15$ for Tal-treated mice. Data were presented as mean values ± SEM. $P$ values were determined by a two-tailed, unpaired, Mann–Whitney test. * at 2 weeks indicates $P = 0.0169$; *** at 3 weeks indicates $P = 0.0007$; ** at 5 weeks indicates $P = 0.0023$, and **** at 7 weeks indicates $P < 0.0001$. (C) B6/129F1 mice were injected with the PARPi-Res, *Brca1*-def breast tumor cells isolated in (B) and then treated with either Veh or Tal in-vivo to test their sensitivity to Tal. $n = 15$ for Veh-treated mice; $n = 30$ for Tal-treated mice. Data were presented as mean values ± SEM. $P$ values were determined by a two-tailed, unpaired, Mann–Whitney test. ns: not significant. (D) Schematic representation of the in-vivo treatment regimen to generate talazoparib-resistant, *Bard1*-def breast tumor cells in mice, which was performed with similar methodologies to those described in (A). (E) A similar experimental protocol was followed as in (A, B) except Tal treatment in mice bearing *Bard1*-def Tal-sensitive ("PARPi-Sen") tumors started at 1 week after tumor-cell injection. *Bard1*-def tumors responded to PARPi until 3 weeks (drug response phase), after which tumors progressed and became drug-refractory. Tal-resistant ("PARPi-Res") tumor cells from drug-refractory tumors were isolated at 5 weeks post tumor injection when mice developed a body-conditioning score (BCS) of 2 or less from cachexia. $n = 5$ for Veh-treated mice; $n = 6$ for Tal-treated mice. Data were presented as mean values ± SEM. $P$ values were determined by a two-tailed, unpaired, Mann–Whitney test. ** at 3 and 4 weeks indicates $P = 0.0043$; ** at 5 weeks indicates $P = 0.0095$. (F) Mice were injected with the PARPi-Res, *Bard1*-def breast tumor cells isolated in (E) and then treated with either Veh or Tal to test their sensitivity to Tal in-vivo. These mouse tumors were refractory to Tal treatment from the onset and were harvested after 2 weeks of treatment as they reached the criteria for euthanasia. $n = 12$ for Veh-treated mice; $n = 14$ for Tal-treated mice. Data were presented as mean values ± SEM. $P$ values were determined by a two-tailed, unpaired, Mann–Whitney test. ns; not significant. Source data are available online for this figure.

compared to 1F), only a modest difference was observed in vitro (EV1A,B), albeit to different extents between the two cell lines. These results suggest that the tumor microenvironment and physiological context may be critical for the development of PARPi resistance in these models.

## PARPi-resistant tumors show increased VEGFR2 and PGF expression but only modest sensitization to PARPi upon VEGFR2 depletion

Since PARPi resistance in the *Brca1*-def and *Bard1*-def models was more faithfully recapitulated in-vivo than in vitro (Figs. 1; EV1), we hypothesized that signals from the tumor microenvironment are important for mediating PARPi resistance. To identify mechanisms of PARPi resistance, we compared the tumor microenvironments of PARPi-sensitive and -resistant tumors by immunohistochemical staining (Figs. 2A–H; EV2A,B). The PARPi-resistant tumors showed a significant increase in CD31$^+$ endothelial cells (Fig. 2A,B), indicating a greater amount of angiogenesis in PARPi-resistant

tumors compared to PARPi-sensitive tumors. Analysis of the major immune-cell populations revealed a marked reduction in the number of CD8$^+$ T cells and increased numbers of both CD4$^+$ helper T cells and CD11C$^+$ dendritic cells in PARPi-resistant compared to PARPi-sensitive tumors (EV2A,B). However, B cells and macrophages showed no consistent differences between the sensitive and resistant tumors in the *Brca1*-def and *Bard1*-def models (EV2A,B). These results indicate that changes in angiogenesis and immune-cell composition may underlie PARPi resistance in *BRCA1*-mutant breast tumors.

Since the vascular endothelial growth factor (VEGF) signaling pathway regulates both angiogenesis and the immune response (Ferrara et al, 2003; Zhang and Brekken, 2022), we performed immunohistochemical staining to determine whether the VEGF pathway is activated in the PARPi-resistant tumors. Among the VEGF family of ligands, VEGFA expression was modestly increased while placental growth factor (PGF) was markedly increased (*Brca1*-def: 5.9-fold, $p = 0.0079$ and *Bard1*-def: 22.5-fold, $p = 0.0159$) in the PARPi-resistant tumors (Fig. 2C–F). These

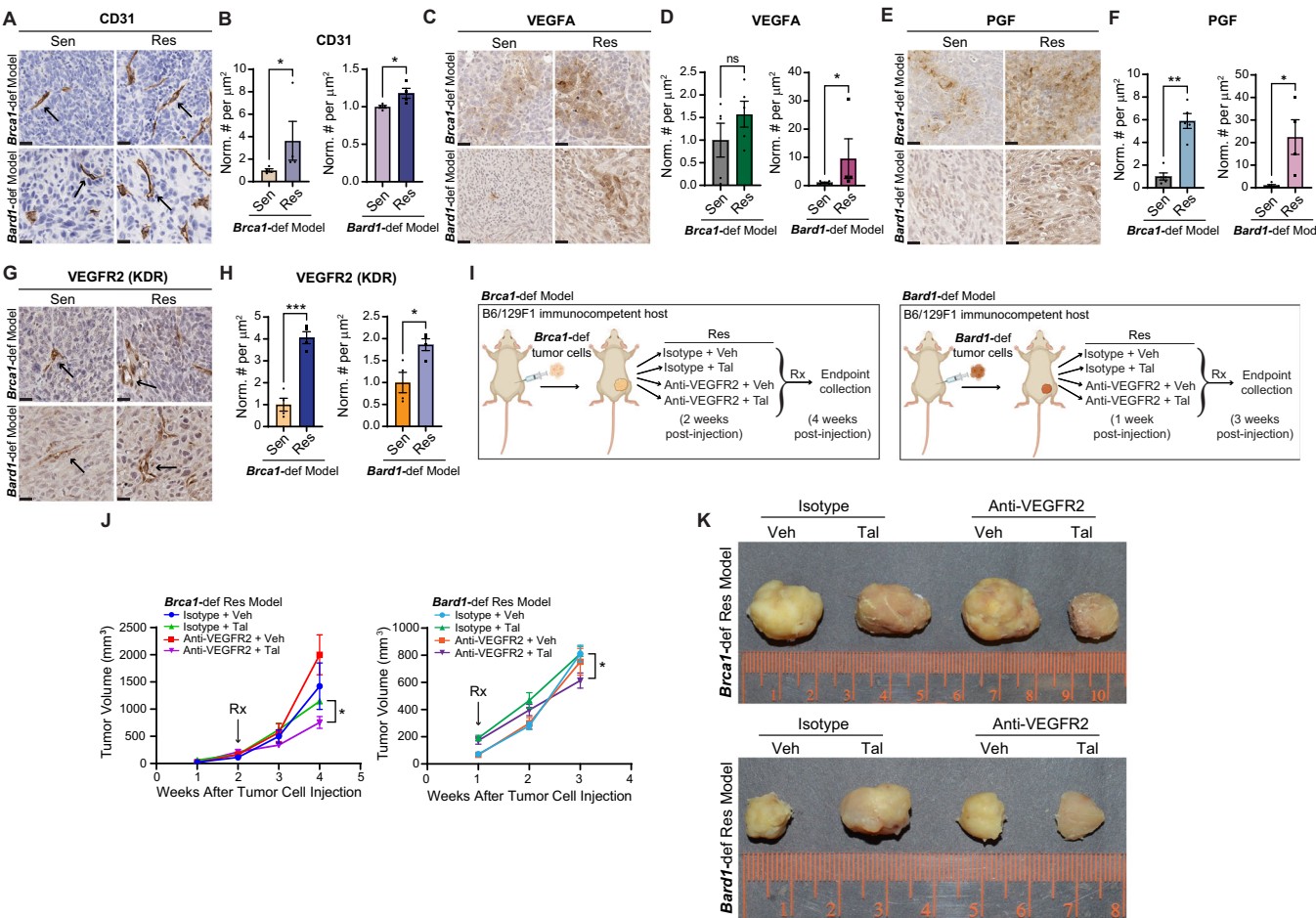

**Figure 2. Talazoparib (Tal)-resistant tumors show increased VEGFR2 (KDR) and PGF expression but only modest sensitization to Tal upon VEGFR2 depletion.**

(A) Representative images of immunohistochemistry (IHC) for CD31 on talazoparib-sensitive ("Sen") and -resistant ("Res") tumors from the *Brca1*-def and *Bard1*-def models described in Fig. 1. Arrows mark a few examples of endothelial cells, identified by both CD31$^+$ immunostaining and morphology. Scale bars, 20 μm. (B) Immunostained tumor sections from (A) were quantified using automated QuPath software to identify positively stained cells. $n = 4$ Sen and Res tumors for both models. Data were presented as mean values ± SEM. $P$ values were determined by a two-tailed, unpaired, Mann–Whitney test. * indicates $P = 0.0286$ for both models. (C–F) Representative images of IHC for VEGFA (C) or PGF (E) on Sen and Res tumor sections from the *Brca1*-def and *Bard1*-def models are described in Fig. 1. Scale bars, 20 μm. Immunostained tumor sections from (C) or (E) were quantified using automated QuPath software to identify positively stained cells (D, F). For both VEGFA and PGF in the *Brca1*-def model, $n = 5$ tumors for both Sen and Res, and for the *Bard1*-def model, $n = 5$ Sen tumors, and $n = 4$ Res tumors. Data were presented as mean values ± SEM. $P$ values were determined by a two-tailed, unpaired, Mann–Whitney test. * indicates $P = 0.0317$ for the *Bard1*-def model (VEGFA) in (D); ns; not significant. For PGF staining in (F), ** indicates $P = 0.0079$ for the *Brca1*-def model, and * indicates $P = 0.0159$ for the *Bard1*-def model. (G) Representative images of IHC for VEGFR2 (KDR) on Sen and Res tumor sections from the *Brca1*-def and *Bard1*-def models described in Fig. 1. Arrows show a few examples of VEGFR2$^+$ endothelial cells. Scale bars, 20 μm. (H) Immunostained tumor sections from (G) were quantified using automated QuPath software to identify positively stained cells. $n = 4$ tumors for both Sen and Res for both models. Data were presented as mean values ± SEM. $P$ values were determined by a two-tailed, unpaired, Welch's *t*-test. *** indicates $P = 0.0003$ for the *Brca1*-def model, and * indicates $P = 0.0249$ for the *Bard1*-def model. (I) Schematic representation of the experimental design to test the effect of in-vivo inhibition of VEGFR2 in Res tumors from the *Brca1*-def and *Bard1*-def models. For the *Brca1*-def model, Res tumor cells were injected in mice, and 2 weeks later, randomized tumor-bearing mice received one of the following four treatments: (1) isotype control antibody + vehicle ("Isotype + Veh"), (2) isotype + Tal, (3) anti-mouse VEGFR2 antibody ("Anti-VEGFR2") + Veh, and (4) anti-VEGFR2 + Tal. *Brca1*-def mice were then euthanized for tumor collection at 4 weeks post Res-cancer-cell injection. For the *Bard1*-def model, mice started receiving treatments at 1 week post Res-cancer-cell injection and were euthanized at 3 weeks post Res-cancer-cell injection. (J) Tumor growth curves for the Res-tumor-bearing mice described in (I). For the *Brca1*-def model, $n = 4$ mice for the Isotype + Veh group, $n = 4$ mice for the Isotype + Tal group, $n = 4$ mice for the Anti-VEGFR2 + Veh group, and $n = 5$ mice for the Anti-VEGFR2 + Tal group. For the *Bard1*-def model, $n = 3$ mice for the Isotype + Veh group, $n = 6$ mice for the Isotype + Tal group, $n = 3$ mice for the Anti-VEGFR2 + Veh group, and $n = 7$ mice for the Anti-VEGFR2 + Tal group. Data were presented as mean values ± SEM. $P$ values were determined by a two-tailed, unpaired, Student's *t*-test, comparing endpoint tumor volumes between the Isotype + Tal and Anti-VEGFR2 + Tal groups. For the *Brca1*-def model, * at 4 weeks indicates $P = 0.0174$. For the *Bard1*-def model, * at 3 weeks indicates $P = 0.0306$. (K) Representative images of tumors following the treatment regimens from (I). Source data are available online for this figure.

results indicate that signaling through the VEGF family of ligands might be important for PARPi resistance.

VEGFR2 (also known as KDR) is a key receptor in vascular endothelial cells that recognizes the VEGF family of ligands and serves as a major signal transducer for angiogenesis (Shibuya,

2001). Consistent with an increase in CD31$^+$ endothelial cells (Fig. 2A,B), the number of VEGFR2-expressing endothelial cells also increased significantly in the PARPi-resistant tumors compared to PARPi-sensitive tumors (Fig. 2G,H). Pathological evaluation of the immunohistochemistry revealed that VEGFR2 is

expressed primarily in the endothelial cells (as expected) but not in the tumor cells (Fig. 2G). Since PARPis can synergize with anti-angiogenic agents in ovarian cancer (Le Saux et al, 2021), we examined whether blocking pro-angiogenic VEGFR2 signaling could sensitize PARPi-resistant breast tumors to talazoparib. To this end, we implanted PARPi-resistant tumor cells from the *Brca1*-def and *Bard1*-def models in the mammary glands of new recipient B6/129F1 mice and treated them with either VEGFR2 antibody (200 µg/mouse three times a week) or isotype control antibody in combination with vehicle or talazoparib (schematic in Fig. 2I). VEGFR2 blockade in combination with talazoparib reduced the number of CD31+ endothelial cells in PARPi-treated tumors in both the *Brca1*-def and *Bard1*-def models (EV2C), which confirmed the efficacy of the VEGFR2 antibody treatment. Interestingly, despite reduced angiogenesis, VEGFR2 inhibition in combination with talazoparib only modestly reduced tumor growth in the *Brca1*-def and *Bard1*-def models (Figs. 2J,K; EV2D), suggesting that other mechanisms are important for driving in-vivo PARPi resistance.

## FLT1 (VEGFR1) activation in tumor cells promotes PARPi resistance in breast cancer

The high expression of PGF in PARPi-resistant tumors (Fig. 2E,F) prompted us to investigate whether the PGF receptor FLT1 (also known as VEGFR1) contributes to PARPi resistance. Indeed, by immunohis-tochemical analysis, we observed a striking increase in levels of both phosphorylated FLT1 (Tyr1213; referred to as "pFLT1" hereafter) and total FLT1 in tumor cells from PARPi-resistant tumors compared to PARPi-sensitive tumors (Figs. 3A,B; EV3A,B). In contrast, no significant changes in FLT4 (also known as VEGFR3) expression were observed (EV3C,D). Based on these observations, we hypothesized that the increased PGF levels in the tumor microenvironment induced by PARPi treatment may, in turn, promote PARPi resistance by activating FLT1 signaling in the tumor cells. To test this hypothesis, we performed loss- and gain-of-function experiments to determine whether genetic repression of *Flt1* in PARPi-resistant tumor cells impacts tumor growth in the presence of PARPi, and whether this effect is rescued by FLT1 re-expression. We first engineered lentiviruses to deplete *Flt1* expression ("*Flt1i*") in the tumor cells by CRISPR-mediated gene repression (CRISPRi) using two independent guide RNAs ("gRNA1" or "gRNA2"), as described previously (Biswas et al, 2022) (EV3E,F). For gain-of-function experiments, we re-expressed FLT1 in FLT1-repressed cancer cells ("*Flt1i + Flt1* o/e", EV3G,H). Next, PARPi-resistant *Brca1*-def and *Bard1*-def tumor cells, transduced with either control lentivirus ("Lenti-Con"), *Flt1i* lentivirus, or *Flt1i* lentivirus plus *Flt1* o/e lentivirus, were injected orthotopically in the mammary glands of syngeneic B6/129F1 mice. As illustrated in Figs. 3C; EV3J, mice were randomized and treated five days/week with either vehicle or talazoparib (0.3 mg/kg body weight/day). An analysis of tumor growth and weight at endpoint revealed that *Flt1* repression in tumor cells re-sensitizes PARPi-resistant tumors to talazoparib (Figs. 3D,E; EV3I), which can be rescued by FLT1 re-expression (EV3K,L).

## A pan-VEGFR antagonist (axitinib) re-sensitizes PARPi-resistant breast tumors to PARPi treatment

We next tested whether PARPi-resistant tumors could also be re-sensitized by blocking FLT1 signaling pharmacologically using the pan-VEGFR inhibitor axitinib, an FDA-approved drug for treating

metastatic renal-cell carcinoma patients (Tyler, 2012). PARPi-resistant *Brca1*-def and *Bard1*-def tumor cells were injected orthotopically into the mammary gland of syngeneic B6/129F1 mice (see schematic in Fig. 4A,D). Mice were randomized and treated five days/week with either vehicle, talazoparib (0.3 mg/kg body weight/day), axitinib (30 mg/kg body weight/day), or talazoparib-plus-axitinib (0.3 mg/kg body weight/day and 30 mg/kg body weight/day, respectively). In contrast to the single-treatment groups, we observed a striking reduction in the tumor burden of mice treated with both talazoparib and axitinib (Figs. 4B,C,E,F; EV4A,B). None of the treatments led to overt toxicities, and stable body weight was maintained by all mice for the duration of the studies (≥3 weeks) (EV4C,D). Thus, the combination of PARPi and FLT1 blockade is highly effective at suppressing PARPi resistance in-vivo and inhibiting the growth of tumors lacking BRCA1 function.

## FLT1 activation induces pro-survival AKT signaling in PARPi-resistant tumor cells

To investigate how FLT1 activation in breast tumor cells counter-acts PARPi-induced cytotoxicity, we first examined whether downstream mediators of the FLT1 pathway are activated in PARPi-resistant tumors. Upon binding to VEGF family ligands (e.g., PGF), FLT1 activates growth and survival pathways, including AKT and STAT3 signaling, in immune and vascular smooth muscle cells (Bartoli et al, 2000; Bellik et al, 2005; Chen et al, 2008; Selvaraj et al, 2003; Tchaikovski et al, 2008). Therefore, PARPi-resistant *Brca1*-def and *Bard1*-def tumor cells expressing either control (Con) or *Flt1*-specific guide RNA (*Flt1i*) were exposed to recombinant PGF in vitro. Notably, we observed FLT1-dependent activation of AKT, but not STAT3, upon PGF treatment (Figs. 5A,B; EV5A,B). Likewise, AKT, but not STAT3, was also activated in PARPi-resistant tumors compared to PARPi-sensitive tumors in-vivo (Figs. 5C,D; EV5C,D). Importantly, the combinations of either talazoparib-plus-*Flt1i* or talazoparib-plus-axitinib in-vivo signifi-cantly suppressed AKT activation in the PARPi-resistant tumors in mice (Fig. 5E–H). These results suggest that FLT1-AKT pathway activation allows breast tumor cells lacking BRCA1 function to escape PARPi-induced cytotoxicity in-vivo.

## The cytotoxic immune response is restored by the combination of PARPi and FLT1 blockade

Based on the known immune-modulatory functions of VEGF signaling (Zhang and Brekken, 2022) and our observation that PARPi-resistant tumors exhibit reduced numbers of CD8+ T cells (EV2A,B), we asked whether inhibition of FLT1 signaling might indirectly impact the numbers of CD8+ T cells, and other immune cells, in PARPi-resistant tumors (Fig. 6). Indeed, increased CD8+ T-cell infiltration was observed in the PARPi-resistant tumors treated with talazoparib in combination with FLT1 blockade, either genetically (Fig. 6A,B) or pharmacologically (Fig. 6F,G). Impor-tantly, the tumor regression observed in B6/129F1 mice upon combined treatment with talazoparib and FLT1 blockade (Figs. 3C–E and EV3I and Figs. 4 and EV4A,B) did not occur in T-cell-deficient nude-*Foxn1nu* mice (Fig. 6C–E and Fig. 6H–J). Of note, CD4+ helper T cells, B220+ B cells and F4/80+ macrophages also showed a trend towards increased tumor infiltration upon FLT1 blockade in the *Brca1*-def and *Bard1*-def models (Appendix

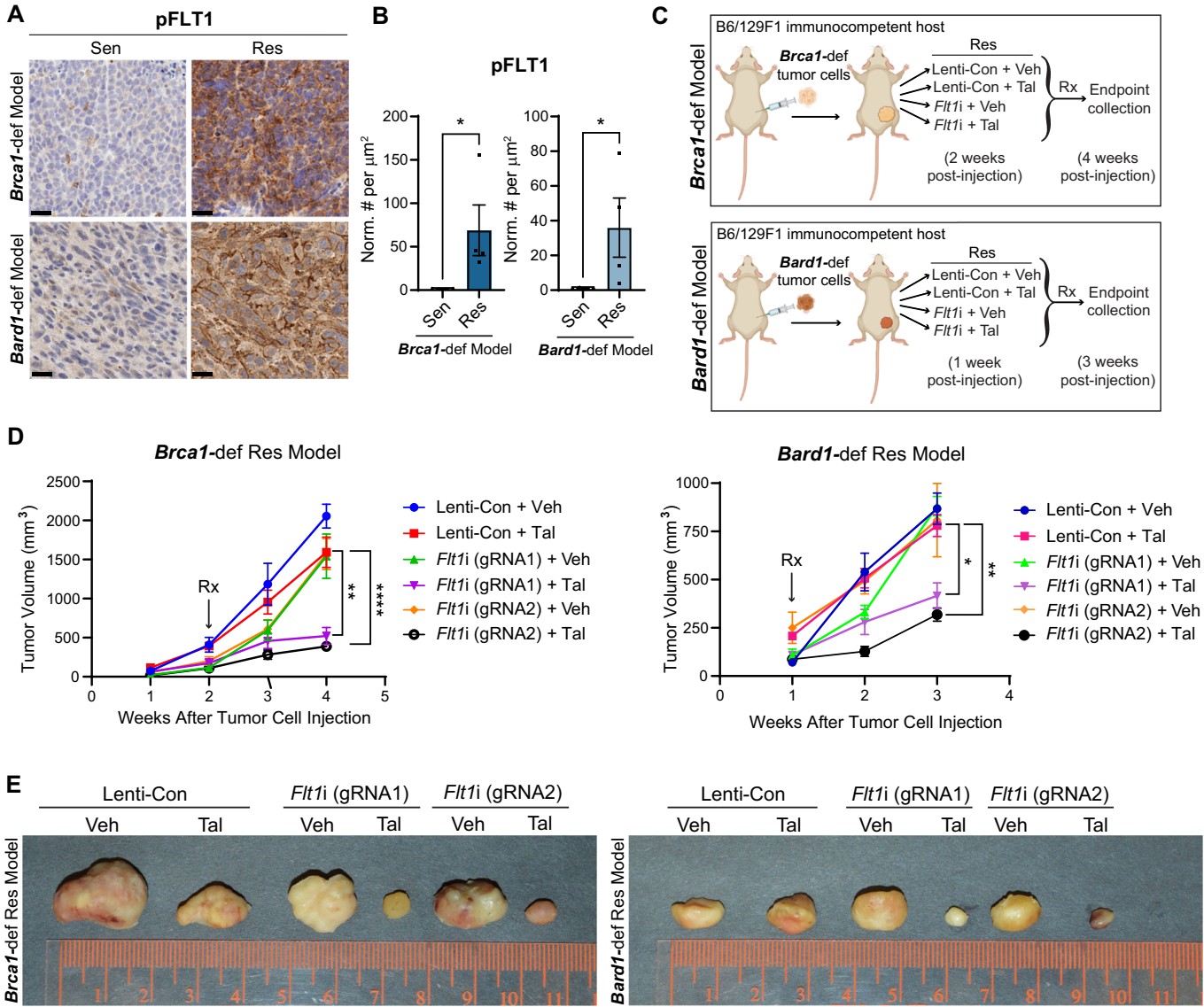

**Figure 3. FLT1 (VEGFR1) activation in tumor cells promotes talazoparib resistance in breast cancer models.**

(**A**) Representative images of IHC for phosphorylated FLT1 (pFLT1) on sections from the talazoparib-sensitive ("Sen") and -resistant ("Res") *Brca1*-def and *Bard1*-def breast cancer tumors from mouse models described in Fig. 1. Scale bars, 20 µm. (**B**) Immunostained sections from (**A**) were quantified using automated QuPath software to identify positively stained cells. $n = 4$ tumors for both Sen and Res in both models. Data were presented as mean values ± SEM. $P$ values were determined by a two-tailed, unpaired, Mann–Whitney test. * indicates $P = 0.0286$ for both models. (**C**) Schematic representation of the experiment designed to test whether *Flt1* is required for the talazoparib-resistance phenotype in *Brca1*-def and *Bard1*-def mammary tumors. Res lines derived from the *Brca1*-def and *Bard1*-def models described in Fig. 1 were transduced with either control lentivirus ("Lenti-Con") or lentivirus encoding guide RNA for *Flt1* ("*Flt1*i") using two independent gRNAs and injected into mice. For the *Brca1*-def model, randomized mice received either vehicle ("Veh") or talazoparib ("Tal") treatment starting at 2 weeks following Res-tumor-cell injection and were euthanized at 4 weeks following injection. For the *Bard1*-def model, randomized mice received treatment at 1 week following Res-tumor-cell injection and were euthanized at 3 weeks following injection. (**D**) Tumor growth curves for the experiment described in (**C**). For the *Brca1*-def model, $n = 6$ for Lenti-Con + Veh, $n = 8$ for Lenti-Con + Tal, $n = 5$ for *Flt1*i (gRNA1) + Veh or Tal and *Flt1*i (gRNA2) + Veh, and $n = 7$ for *Flt1*i (gRNA2) + Tal treatment groups. For the *Bard1*-def model, $n = 5$ for Lenti-Con + Veh or Tal, *Flt1*i (gRNA1) + Veh or Tal, and *Flt1*i (gRNA2) + Tal, and $n = 3$ for *Flt1*i (gRNA2) + Veh. Data were presented as mean values ± SEM. $P$ values were determined with a one-way ANOVA test, comparing endpoint tumor volumes between the Lenti-Con + Tal and *Flt1*i (gRNA1 and gRNA2) + Tal groups. For the *Brca1*-def model at 4 weeks, ** between Lenti-Con + Tal and *Flt1*i (gRNA1) + Tal indicates $P = 0.0017$ and **** between Lenti-Con + Tal and *Flt1*i (gRNA2) + Tal indicates $P < 0.0001$. For the *Bard1*-def model at 3 weeks, * between Lenti-Con + Tal and *Flt1*i (gRNA1) + Tal indicates $P = 0.0190$ and ** between Lenti-Con + Tal and *Flt1*i (gRNA2) + Tal indicates $P = 0.0021$. (**E**) Representative images of tumors at endpoint are described in (**D**). Source data are available online for this figure.

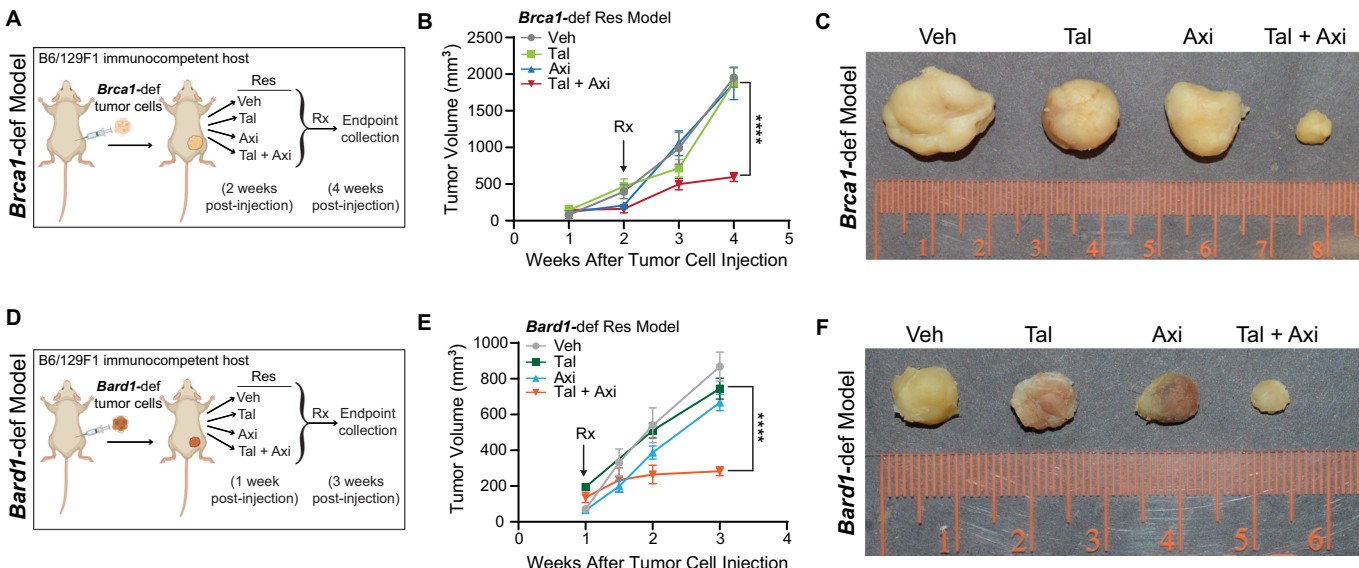

**Figure 4. Pharmacological inhibition of FLT1 re-sensitizes talazoparib-resistant tumors to talazoparib treatment.**

(A) Schematic representation of the experiment designed to test whether FLT1 contributes to Tal-PARPi resistance. Mice were injected with the Tal-resistant ("Res") *Brca1*-def breast tumor cells from Fig. 1 and then randomized into the following four treatment groups at 2 weeks post tumor-cell injection: (1) vehicle ("Veh"), (2) talazoparib ("Tal"), (3) axitinib ("Axi"), and (4) Tal + Axi. Tumor size was measured weekly to monitor tumor growth, and mice were euthanized for tumor collection at 4 weeks post tumor-cell injection. (B) Tumor growth curves for the mice treated as described in (A). $n = 6$ Veh-treated tumors, $n = 7$ Tal-treated tumors, $n = 5$ Axi-treated tumors, and $n = 5$ tumors treated with Tal + Axi. Data were presented as mean values ± SEM. *P* values were determined with a one-way ANOVA test, comparing endpoint tumor volumes between the Tal and Tal + Axi groups. **** at 4 weeks indicates $P < 0.0001$. (C) Representative images of tumors from (B). (D) Schematic representation to test whether FLT1 inhibition re-sensitizes talazoparib-resistant *Bard1*-def tumors to talazoparib using the same methodologies described in (A). Treatment started at 1 week post Res-tumor-cell injection, and tumors were harvested at 3 weeks post tumor-cell injection. (E) Tumor growth curves for tumors in (D). $n = 5$ Veh-treated tumors, $n = 6$ Tal-treated tumors, $n = 5$ Axi-treated tumors, and $n = 7$ tumors treated with Tal + Axi. Data were presented as mean values ± SEM. *P* values were determined with a one-way ANOVA test comparing endpoint tumor volumes between the Tal and Tal + Axi groups. **** at 3 weeks indicates $P < 0.0001$. (F) Representative images of tumors from (E). Source data are available online for this figure.

Fig. S1A,B). These results demonstrate that re-sensitization of PARPi-resistant tumors by FLT1 blockade is a T-cell-dependent process, which may entail multiple changes in the immune status of the tumor microenvironment.

## Association between FLT1 activation in human tumor cells at pre-treatment and faster progression on PARP inhibitors in breast cancer patients

To validate our preclinical findings, we performed pFLT1 and FLT1 immunostaining on tissue specimens that were obtained prior to PARPi treatment from 10 breast cancer patients harboring mutations in the *BRCA1/2* or *PALB2* DNA damage response (DDR) genes (Fig. 7; Appendix Fig S2A,B; and Appendix Table S1). The immunostained samples were scored as either high or low expression for FLT1 and pFLT1 (activation) in tumor cells. Consistent with our preclinical observations (Figs. 3 and EV3), a blinded pathological examination revealed a statistically significant association between FLT1 expression (both activated and total) at pre-treatment and shorter progression-free survival ($p = 0.012$ for pFLT1 and $p = 0.005$ for total FLT1, Fig. 7A–C; Appendix Fig. S2A,B) in patients with breast cancer. These findings suggest that FLT1 activation in human breast tumor cells (pre-treatment) is significantly associated with a higher risk of progression on PARPi, and thus pFLT1/FLT1 status in human breast tumors with *BRCA1/2* or *PALB2* mutations could serve as a biomarker to stratify

patients for benefit from combination treatment with PARPi and FLT1 blockade.

## Discussion

The emergence of drug resistance is a major clinical challenge that limits the efficacy and durability of PARPi therapies for *BRCA1/2*-mutant breast cancer patients (Tung and Garber, 2022). Although PARPi treatment elicits 60% response rates and longer PFS compared to conventional chemotherapy agents (Litton et al, 2018; Robson et al, 2017), it fails to improve overall survival due to the onset of drug resistance (D'Andrea, 2018; Robson et al, 2017). Thus, overcoming drug resistance should improve the efficacy of PARPi treatment and extend the survival of breast cancer patients. Here, we demonstrate that FLT1 signaling can potentially serve as a biomarker and therapeutic target to circumvent PARPi resistance in breast cancer patients.

In this study, we describe new mouse models, based on *Brca1*-def and *Bard1*-def orthotopic allografts, that recapitulate the clinical response and progression phases of PARPi therapy observed in *BRCA1/2*-mutant breast cancer patients. In contrast to the well-established PARPi-resistance mechanisms based on restoration of BRCA1/2 pathway functions, we identify an adaptive mechanism driven by PGF-FLT1-AKT signaling that protects *Brca1*- and *Bard1*-deficient breast tumor cells from PARPi-induced

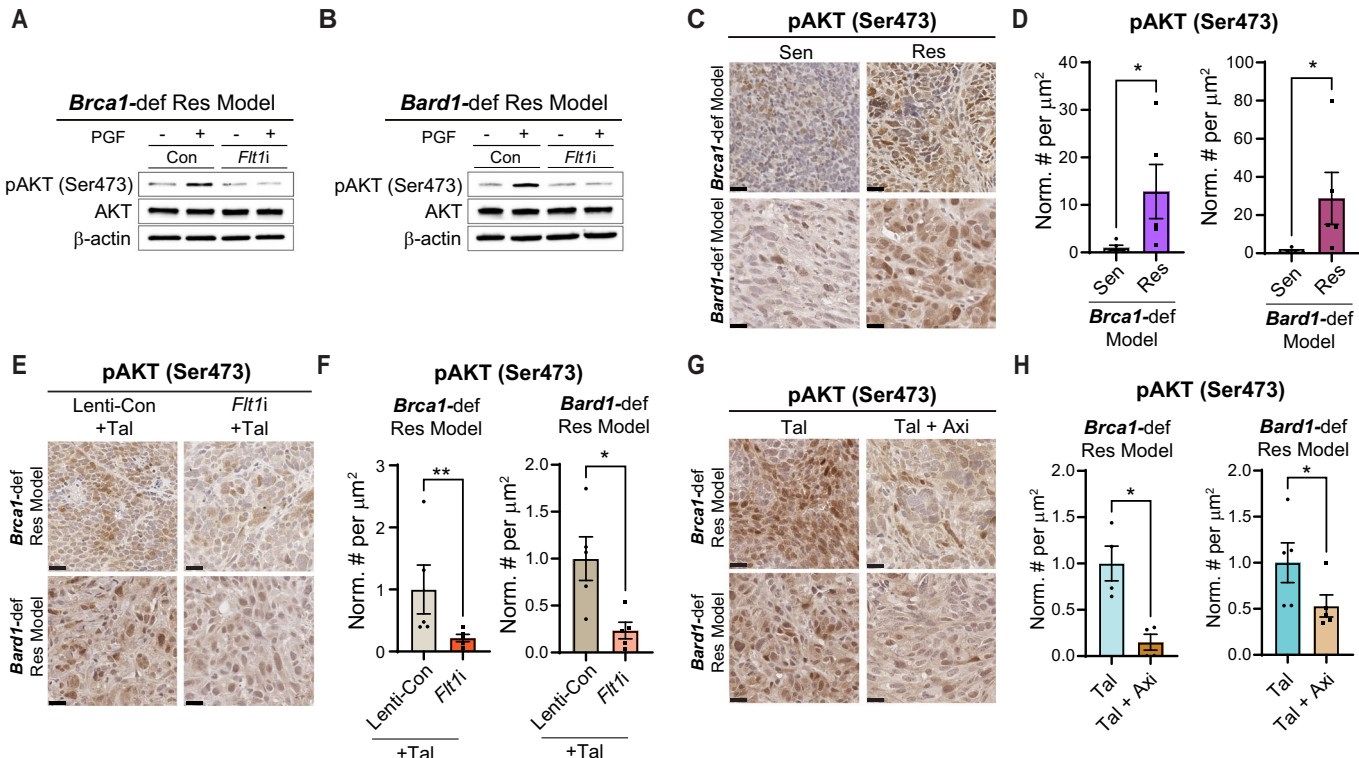

**Figure 5. FLT1 activation in *Brca1*- and *Bard1*-def breast tumor cells induces pro-survival AKT signaling.**

(A, B) Immunoblot analysis was performed on lysates from *Flt1*-expressing (Con) and -deficient (*Flt1*i), talazoparib-resistant ("Res") *Brca1*- and *Bard1*-def tumor cells, that were treated with 50 ng/ mL of mouse PGF protein using antibodies against phosphorylated-AKT at serine 473 (pAKT Ser473), AKT, and β-actin. (C) Representative images of IHC for pAKT Ser473 staining on talazoparib-sensitive ("Sen") and -Res tumors from the *Brca1*-def and *Bard1*-def breast cancer models from Fig. 1. Scale bars, 20 μm. (D) Immunostained sections from (C) were quantified using automated QuPath software to identify positively stained cells. *n* = 5 mice for Sen and Res groups for both models. Data were presented as mean values ± SEM. *P* values were determined by a two-tailed, unpaired, Mann–Whitney test: * indicates *P* = 0.0159 for both models. (E) Representative images of IHC for pAKT Ser473 staining on talazoparib-treated "Tal", Lenti-Con- and *Flt1*i-expressing Res tumor sections from both models (see Fig. 3C). Scale bars, 20 μm. (F) Immunostained sections from (E) were quantified using automated QuPath software to identify positively stained cells. *n* = 5 mice for both Lenti-Con + Tal and *Flt1*i + Tal groups for the *Brca1*-def and *Bard1*-def models. Data are presented as mean values ± SEM. *P* values were determined by a two-tailed, unpaired, Mann–Whitney test. ** indicates *P* = 0.0079 for the *Brca1*-def model, and * indicates *P* = 0.0159 for the *Bard1*-def model. (G) Representative images of IHC for pAKT Ser473 on Tal-treated and Tal + Axi-treated tumor sections from both models (see Fig. 4A/D). Scale bars, 20 μm. (H) Immunostained sections from (G) were quantified using automated QuPath software to identify positively stained cells. For the *Brca1*-def model, *n* = 4 tumors from both Tal-treated and Tal + Axi-treated mice. For the *Bard1*-def model, *n* = 5 tumors from both Tal-treated and Tal + Axi-treated mice. Data were presented as mean values ± SEM. *P* values were determined by a two-tailed, unpaired, Mann–Whitney test. For the *Brca1*-def model, * indicates *P* = 0.0286 and for the *Bard1*-def model, * indicates *P* = 0.0317. Source data are available online for this figure.

cell death in-vivo. Since PGF expression increases locally in the tumor milieu upon PARPi treatment (Fig. 2), it is possible that a minor subpopulation of *Brca1*- and *Bard1*-deficient tumor cells that express the PGF receptor FLT1 prior to PARPi treatment becomes enriched in the PARPi-resistant tumors. Therefore, the FLT1 pathway represents a vulnerability that can be targeted to overcome PARPi resistance. As such, our preclinical studies might offer a new biomarker-guided combination treatment option that includes a PARPi (e.g., talazoparib) and a VEGFR inhibitor (e.g., axitinib) to specifically target PARPi-resistant breast cancers that express FLT1.

The function of FLT1 varies by cell type and cellular context. *Flt1* is normally expressed in endothelial cells, immune cells, and hematopoietic stem cells (HSCs) (Ferrara et al, 2003). In the context of angiogenesis, *Flt1* is essential for the organization of embryonic vasculature but not for endothelial cell differentiation (Fong et al, 1995; Fong et al, 1999). FLT1 also functions as a signaling receptor in myeloid cells and HSCs by promoting their

chemotaxis and migration in response to VEGF and/or PGF (Barleon et al, 1996; Clauss et al, 1996; Hattori et al, 2002; Hiratsuka et al, 1998). In the context of cancer, *Flt1*+ bone-marrow-derived hematopoietic progenitor cells promote the formation of pre-metastatic clusters and enhance tumor metastasis in mice (Kaplan et al, 2005). In addition, FLT1 signaling in macrophages activates an inflammatory response and promotes breast cancer metastasis (Qian et al, 2015). Here, we show that FLT1 activation in tumor cells is also clinically relevant, in this case by promoting PARPi resistance in breast cancer through a combination of cell-intrinsic and -extrinsic pathways.

FLT1 expression has been previously reported in a subset of tumor cells where it promotes tumor growth by activating mitogenic pathways (Frank et al, 2011; Lesslie et al, 2006; Sopo et al, 2019; Wey et al, 2005; Wu et al, 2006; Yao et al, 2011). In line with these observations, the efficacy of anti-PGF antibodies strongly correlates with the expression of tumor-derived FLT1

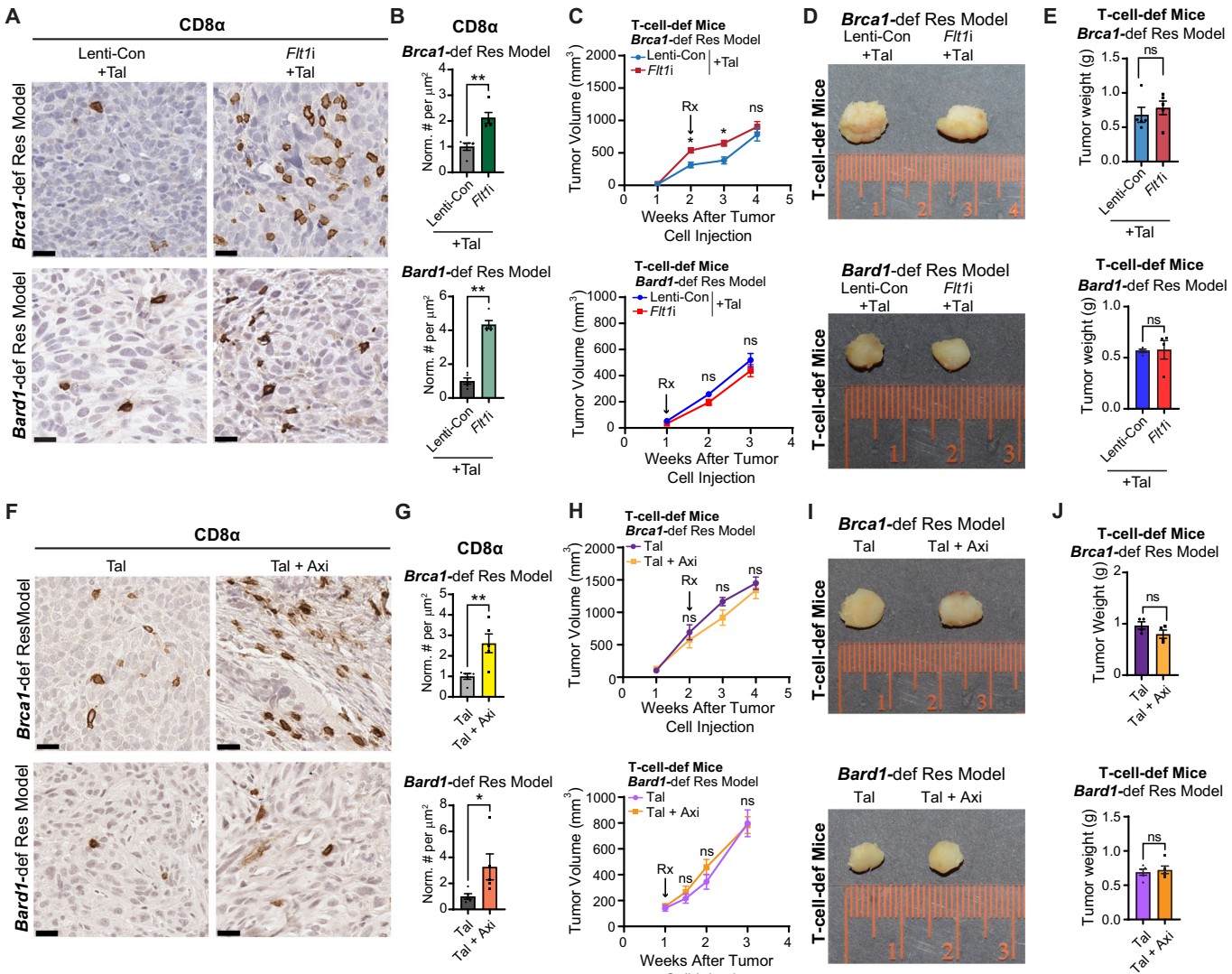

but not with the inhibition of angiogenesis (Yao et al, 2011). This is consistent with our finding that FLT1 activation in the tumor cells represents a key in-vivo determinant of PARPi resistance in our breast cancer models. However, one point of distinction between our current study and the published literature is that *Flt1*-proficient and -deficient tumor cells show only modest growth differences in the absence of PARPi treatment in our tumor models. The difference in tumor growth only becomes pronounced after PARPi treatment, presumably when *Flt1*-deficient tumor cells are eliminated, and *Flt1*-proficient cells persist. We, therefore, hypothesize that PARPi treatment leads to increased PGF expression in tumors, which in turn facilitates pro-survival signaling in FLT1-proficient (but not FLT1-deficient) tumor cells and culminates in the development of PARPi resistance.

In healthy, non-tumor-bearing mice, PARPi treatment reduces angiogenesis in response to growth factors (such as VEGF or PGF) in matrigel plug assays (Tentori et al, 2007). Interestingly, we observed that PARPi causes increased numbers of CD31+ blood vessels and heightened VEGFR2 expression within PARPi-resistant tumors. In addition, VEGFR2 inhibition in combination with

talazoparib only modestly reduced tumor growth in the *Brca1*-def and *Bard1*-def models, whereas *Flt1* inhibition in combination with talazoparib resulted in prominent tumor regression. These data suggest that, in contrast to ovarian cancer models (Bizzaro et al, 2021), VEGFR2-induced angiogenesis does not appear to be a critical component of the growth and survival of PARPi-resistant breast cancer models.

Our genetic experiments with FIt1 inhibition in cancer cells and the depletion experiments with VEGFR2 antibody allowed us to differentiate between the contribution of FLT1 and VEGFR2 to PARPi resistance in breast cancer. Our studies have revealed a novel function for FLT1 in promoting PARPi resistance. These findings are clinically relevant since the FLT1 pathway could serve as both biomarker and therapeutic target for overcoming PARPi resistance in breast cancer.

Our data also suggest that the FLT1 pathway promotes PARPi resistance, at least in part, by suppressing a cytotoxic immune response in the tumor microenvironment. In particular, we observed reduced numbers of CD8+ T cells in PARPi-resistant, but not PARPi-sensitive, tumors from the *Brca1*- and *Bard1*-deficient tumor

**Figure 6. Tumor-regression of *Brca1*-def and *Bard1*-def breast tumors with PARPi and FLT1 blockade is T-cell-dependent.**

(A) Representative images of IHC for CD8α on Lenti-Con- and *Flt1*i-expressing talazoparib-resistant ("Res") tumor sections from the *Brca1*-def and *Bard1*-def breast cancer models (see Fig. 3C). Scale bars, 20 μm. (B) Immunostained sections from (A) were quantified using automated QuPath software to identify positively stained cells. $n = 5$ tumors for both Lenti-Con + Tal and *Flt1*i + Tal in both models. Data were presented as mean values ± SEM. *P* values were determined by a two-tailed, unpaired, Mann–Whitney test. ** indicates $P = 0.0079$ for both models. (C) Cells derived from Lenti-Con- and *Flt1*i-expressing *Brca1*-def and *Bard1*-def Res tumors (see Fig. 3C) were injected into athymic nu/nu strain immunodeficient mice ("T-cell-def mice" in the rest of the figure). Following Res-tumor-cell injection, both Lenti-Con and *Flt1*i groups were treated with Tal. For the *Brca1*-def model, treatment started at 2 weeks post tumor-cell injection, and tumors were collected at 4 weeks post tumor-cell injection. For the *Bard1*-def model, treatment started at 1 week post tumor-cell injection, and tumors were collected at 3 weeks post tumor-cell injection. Tumor size was measured weekly. For the *Brca1*-def model, $n = 5$ mice for both Lenti-Con + Tal and *Flt1*i + Tal groups. For the *Bard1*-def model, $n = 4$ mice for both Lenti-Con + Tal and *Flt1*i + Tal groups. Data were presented as mean values ± SEM. *P* values were determined by a two-tailed, unpaired, Mann–Whitney test. For the *Brca1*-def model, * at 2 weeks indicates $P = 0.0317$ and * at 3 weeks indicates $P = 0.0159$; ns: not significant. (D) Representative images of tumors from the experiment described in (C). (E) Tumor weights were plotted at endpoint. For the *Brca1*-def model, $n = 5$ tumors for both Lenti-Con + Tal and *Flt1*i + Tal. For the *Bard1*-def model, $n = 4$ tumors for both Lenti-Con + Tal and *Flt1*i + Tal. Data were presented as mean values ± SEM. *P* values were determined by a two-tailed, unpaired, Mann–Whitney test. ns; not significant. (F) Representative images of IHC for CD8α on Res tumor sections from the mice treated with either Tal or Tal + Axi (see Fig. 4A/D). Scale bars, 20 μm. (G) Immunostained sections from (F) were quantified using automated QuPath software to identify positively stained cells. $n = 5$ mice for both Tal and Tal + Axi groups from both models. Data were presented as mean values ± SEM. *P* values were determined by a two-tailed, unpaired, Mann–Whitney test. For the *Brca1*-def model, ** indicates $P = 0.0079$. For the *Bard1*-def model, * indicates $P = 0.0159$. (H) *Brca1*-def and *Bard1*-def Res tumor cells were injected into T-cell-def mice. Mice were then randomized to receive either Tal or Tal + Axi treatments. For the *Brca1*-def model, treatment started at 2 weeks post tumor-cell injection, and tumors were collected at 4 weeks post tumor-cell injection. For the *Bard1*-def model, treatment started at 1 week post tumor-cell injection, and tumors were collected at 3 weeks post tumor-cell injection. For the *Brca1*-def model, $n = 4$ for both Tal and Tal + Axi groups. For the *Bard1*-def model, $n = 5$ for both Tal and Tal + Axi groups. Data were presented as mean values ± SEM. *P* values were determined by a two-tailed, unpaired, Mann–Whitney test. ns; not significant. (I) Representative images of tumors treated with either Tal or Tal + Axi (from H) for both models. (J) Tumor weights were plotted at endpoint. For the *Brca1*-def model, $n = 4$ tumors for both Tal and Tal + Axi groups. For the *Bard1*-def model, $n = 5$ tumors for both Tal and Tal + Axi groups. Data were presented as mean values ± SEM. *P* values were determined by a two-tailed, unpaired, Mann–Whitney test. ns; not significant. Source data are available online for this figure.

**A**

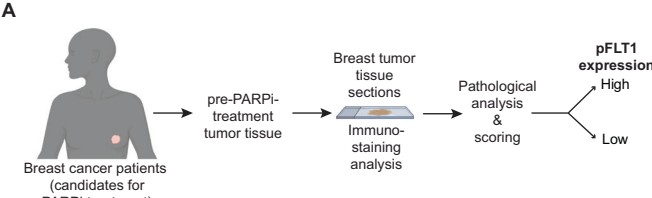

**B**

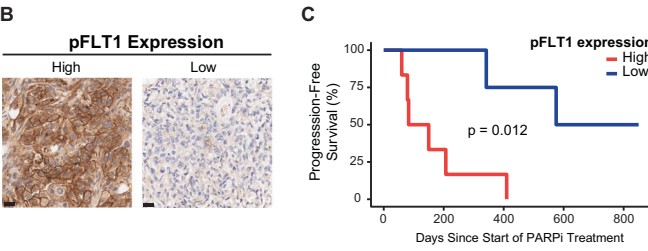

**Figure 7. High pFLT1 expression in human breast tumors prior to PARPi treatment is associated with shorter progression-free survival on PARPi in patients with breast cancer.**

(A) Schematic representation of the workflow for the pathological evaluation of pFLT1 expression in tumor cells from breast cancer patients before PARPi treatment. pFLT1 immunostainings were performed on tissue specimens (biopsies/resected material) from ten patients with breast cancer and were obtained prior to PARPi treatment. The immunostained samples were scored by a pathologist, who was blinded to the sample details, as either pFLT1-high or -low expression. (B) Representative IHC images of high or low levels of pFLT1 expression on tumor tissue samples from patients with breast cancers. Scale bars, 20 μm. (C) Kaplan–Meier plots for the PFS of patients described in (A). Data were analyzed using the log-rank test: $\chi^2 = 6.325$, degrees of freedom (d.f.) = 1; $P = 0.012$; $n = 10$ patients. Source data are available online for this figure.

models. Moreover, while FLT1 blockade with PARPi treatment is accompanied by increased CD8+ T-cell infiltration and tumor regression in immunocompetent mice, these effects were absent in T-cell-deficient nude-*Foxn1*[nu] mice. Our findings, therefore, suggest

that FLT1 inhibition counters PARPi resistance through at least two distinct mechanisms. First, FLT1 blockade interrupts pro-survival PGF-FLT1-AKT signaling in tumor cells, and second, it increases T-cell infiltration and the immune response, possibly by altering the tumor-cell secretome and chemokine milieu. It is currently unclear how FLT1 inhibition in tumor cells alters the secretome and impacts T-cell numbers and function. Future studies will be needed to determine whether the suppression of T-cell cytotoxicity results from decreased chemotaxis, proliferation, or immunosuppression, or through interactions with other immunosuppressive cells in the tumor microenvironment. Although the PGF-FLT1 axis in tumor cells has not been investigated in the context of adaptive immunity, PGF secretion has been linked to immunosuppression (Albonici et al, 2019; Incio et al, 2016). For instance, PGF can induce dendritic-cell dysfunction and suppression of naïve CD4+ T-cell proliferation, thereby skewing T-cell responses toward Th2 (Lin et al, 2007). PGF can also immunosuppress CD8+ T cells by macrophage polarization (Albonici et al, 2019). It is possible that increased PGF expression after PARPi treatment reprograms immune cells to maintain an immunosuppressive tumor microenvironment, which can be further exacerbated by FLT1 signaling in tumor cells.

In addition to *BRCA1/2*-mutant breast cancer, PARPi is also approved for the treatment of ovarian, pancreatic, and prostate cancer. Interestingly, tumor-cell expression of *FLT1* has been reported in ovarian, pancreatic, and prostate tumors (Boocock et al, 1995; Tsourlakis et al, 2015; Wey et al, 2005), suggesting that FLT1 signaling could potentially drive PARPi resistance in these cancers as well. PARPi agents have been approved to treat ovarian cancer patients with and without *BRCA1/2* mutations and/or other HR deficiencies, and combined treatment with the PARPi olaparib and the VEGFA-selective blocker bevacizumab is FDA-approved for ovarian cancer as maintenance therapy (Le Saux et al, 2021; O'Malley et al, 2023). Importantly, the combination of olaparib and the pan-VEGFR inhibitor cediranib led to a significantly longer median progression-free survival (PFS,16.5 vs. 8.2 months, hazard ratio of 0.50, $p = 0.007$) compared to olaparib alone in a randomized Phase II

study of relapsed high-grade ovarian cancer patients (Ivy et al, 2016; Liu et al, 2019). In a randomized 1:1:1 Phase III study in ovarian cancer patients comparing (1) cediranib-plus-olaparib, (2) olaparib, and (3) platinum chemotherapy (the standard of care), the treatment with cediranib-plus-olaparib showed similar clinical activity to platinum in the *BRCA1/2*-mutation-positive group (Liu et al, 2022). However, in the *BRCA1/2*-wild-type group, cediranib-plus-olaparib significantly prolonged PFS compared to olaparib alone in both trials, suggesting that alternative modes of action for this combination treatment exist beyond HR-dependent synthetic lethality. From clinical trial data, it will be interesting to retrospectively determine whether PARPi treatment activates PGF-FLT1 signaling in human ovarian tumor cells (similar to breast cancer models), which, when effectively blocked by cediranib-plus-olaparib, translates to clinical responses independent of *BRCA1/2* status.

In our retrospective analyses of human breast tumors with *BRCA1/2* or *PALB2* mutations from patients, we observed a strong association between FLT1 activation in human tumor cells at pretreatment and subsequent development of PARPi resistance. Although our sample size is small, these findings lay the foundation for future biomarker studies to test whether FLT1 activation in pretreatment biopsies can be used to stratify breast cancer patients at high risk for rapidly acquiring resistance to PARPi. Importantly, these patients might clinically benefit from the combined inhibition of PARP and FLT1. Our study also highlights the need for testing non-genetic markers of PARPi resistance in patient samples by, for example, immunohistochemical staining for pFLT1 and FLT1 in tumor tissues pre- and post-treatment. Contrary to the notion that angiogenesis facilitates PARPi resistance (Alvarez Secord et al, 2021), our findings suggest that blocking angiogenic pathways driven by VEGFA or VEGFR2 alone might have only minimal impact on PARPi resistance. Instead, agents that disrupt the PGF/FLT1/AKT pathway, such as the pan-VEGFR inhibitor axitinib, are more likely to be effective at re-sensitizing PARPi-resistant breast cancers to PARPi. Since axitinib is already FDA-approved for metastatic renal-cell carcinoma (Tyler, 2012) and is currently being tested in combination with talazoparib in a Phase Ib/II clinical trial (NCT04337970) in renal cancer, our findings provide the rationale for repurposing this combination treatment in breast cancer patients with mutations in *BRCA1/2* or *PALB2*. Moreover, if PGF-FLT1 signaling is increased in PARPi-resistant human breast cancers, future studies may validate the use of anti-PGF antibodies as another opportunity for therapeutic intervention to overcome PARPi resistance. This is particularly encouraging with renewed interest in TB-403, a monoclonal PGF-blocking antibody (Lassen et al, 2012; Martinsson-Niskanen et al, 2011) that showed promise in a Phase I clinical trial for medulloblastoma patients (Saulnier-Sholler et al, 2022). Collectively, our studies identify a previously unexplored role for FLT1 signaling in driving PARPi resistance in *BRCA1/2*-mutant breast cancers and suggest that pharmacological inactivation of FLT1 could greatly enhance the efficacy of PARPi treatment in patients.

# Methods

## Animal studies

The treatment of mice in this study was conducted in compliance with the ethical regulations and guidelines set forth by the Columbia University Institutional Animal Care and Use Committee (IACUC), the U.S. National Research Council's Guide for the Care and Use of Laboratory Animals, and the US Public Health Service's Policy on Humane Care and Use of Laboratory Animals. The Institutional guidelines of Columbia University Medical Center (CUMC) Institute of Comparative Medicine were followed in these studies under approved protocol AABU4655. Mice were maintained in the CUMC barrier facility under conventional conditions with constant temperature and humidity and fed a standard diet (Labdiet 5053). The *Brca1*-def and *Bard1*-def cells derived from *Brca1*- or *Bard1*-conditionally deleted mice (Shakya et al, 2008) were authenticated, and the loss of *Brca1* and *Bard1* was confirmed by quantitative PCR analysis. These tumor cells were engineered to express luciferase for bioluminescence imaging and the hygromycin antibiotic resistance gene. Female B6129SF1/J mice (purchased from the Jackson Laboratory) and athymic nude-$Foxn1^{nu}$ mice both aged 8-to-9 weeks (purchased from Envigo) were used in this study. These mice were injected with $5 \times 10^5$ *Brca1*-def or *Bard1*-def tumor cells or their derivatives that were never exposed to talazoparib. *Brca1*-def and *Bard1*-def tumor cells (low passage number) were then injected into the mammary gland (orthotopic implantation), as previously described (Acharyya et al, 2012). Tumor growth was monitored weekly by using an electric caliper to measure the length and width of the tumors in millimeters. The tumor volume can be calculated using the formula (length × (width²))/2, where length and width represent the longest and shortest dimensions of the tumor, respectively. Mice were weighed weekly, monitored twice a week, and euthanized in accordance with the IACUC guidelines from Columbia University. The criteria for prompt euthanasia include weight loss of 20% or more, body-conditioning score (BCS) of 2 or less, signs of hunched posture from cachexia, impaired locomotion, or respiratory distress. Euthanasia was conducted by carbon dioxide inhalation with a secondary method of cervical dislocation. Timed collection of tumors was performed in matching cohorts and has been described in the relevant figure legends.

To generate *Brca1*-def and *Bard1*-def talazoparib-resistant tumor cells in-vivo, mice bearing *Brca1*-def and *Bard1*-def tumors were randomized and treated with PARPi. The PARPi talazoparib (Selleckchem) was solubilized in *N,N*-dimethylacetamide (Millipore), and then diluted in 6% Kolliphor® HS 15 (vehicle). Long-term talazoparib treatment via oral gavage was initiated at 0.3 mg/kg/day and administered five days a week. For the *Brca1*-def model, treatment started at 2 weeks post tumor-cell injection, and talazoparib-resistant tumors were collected 13 weeks after tumor-cell injection. For the *Bard1*-def model, treatment started at 1 week post tumor-cell injection, and talazoparib-resistant tumor cells were collected 5 weeks after injection. To isolate tumor cells, tumors were enzymatically dissociated using Dispase II (1 unit/mL, Roche) and collagenase Type I (2 mg/mL, Worthington). Non-tumor cells were eliminated by supplementing the culture medium with 200 μg/mL of hygromycin. For histological analysis, tumors were fixed in 4% paraformaldehyde in PBS for 24 h at 4 °C, washed, and subsequently processed for histology.

For experiments involving antibody and drug treatments, $5 \times 10^5$ talazoparib-resistant tumor cells (for both *Brca1*-def and *Bard1*-def models) were injected into the mammary gland of individual B6129SF1/J mice as described above. For in-vivo VEGFR2 (KDR) inhibition, the anti-VEGFR2 (BE0060, BioXCell) and the isotype

control (BE0088, BioXCell) antibodies were diluted in a buffer (IP0070, BioXCell). Mice were randomly assigned to receive either 200 µg of VEGFR2 antibody or control antibody via intraperitoneal injection twice a week. Mice were euthanized at 4 weeks post tumor-cell injection for the *Brca1*-def model and at 3 weeks post tumor-cell injection for the *Bard1*-def model. Tumors were collected at the endpoint and processed for histology.

For genetic experiments involving *Flt1* inhibition and *Flt1* re-expression in talazoparib-resistant tumor cells in mice, talazoparib was administered as described above. Treatment started at 2 weeks (*Brca1*-def model) or 1 week (*Bard1*-def model) post tumor-cell injection. Mice were euthanized at their respective end points: at 4 weeks post tumor-cell injection for the *Brca1*-def model and at 3 weeks post tumor-cell injection for the *Bard1*-def model. Tumors were collected and subsequently fixed and processed as described above.

For VEGFR pharmacological inhibition studies, mice bearing talazoparib-resistant tumors from *Brca1*-def and *Bard1*-def models were randomized into multiple treatment groups. Talazoparib was solubilized as described previously. Axitinib (Selleckchem) was solubilized in 0.5% carboxymethylcellulose (w/v%). Drugs were administered in mice by oral gavage 5 days a week with a dose of 0.3 mg/kg/day of talazoparib and/or 30 mg/kg/day of axitinib. Treatments started at 2 weeks following tumor-cell injection for the *Brca1*-def model, and at 1 week following tumor-cell injection for the *Bard1*-def model. *Brca1*-def tumors were collected at 4 weeks post tumor-cell injection, and *Bard1*-def tumors were collected at 3 weeks post tumor-cell injection. Tumors were collected, subsequently fixed, and processed for histology.

### Immunostaining analysis

Paraffin-embedded tumors from mice were sectioned at 5-µm thickness. Multiple sections from different depths of the tumors were used for immunostaining analysis. Slides were baked at 60 °C for 1 h and deparaffinized, rehydrated, and treated with 1% hydrogen peroxide for 10 min. Antigen retrieval was performed using either pH 6.0 citrate buffer (Vector Laboratories) or pH 9.0 Tris-based buffer (Vector Laboratories) in a steamer for 30 min, and endogenous avidin and biotin were blocked using avidin- and biotin-blocking reagents (Vector Laboratories), respectively. The slides were further blocked with BSA and goat or rabbit serum (depending on the species of the secondary antibodies), and tissue sections were incubated with primary antibodies, including antibodies against phospho-AKT (S473) (1:100, #4060, Cell Signaling Technology), KDR/VEGFR2 (1:2000, #9698, Cell Signaling Technology), VEGFA (1:300, #AF-493-NA, R&D Systems), PGF (1:300, AF465, R&D Systems), CD8α (1:200, #98941, Cell Signaling Technology), F4/80 (1:500, #70076, Cell Signaling Technology), B220 (1:400, # 553085, BD Pharmingen), CD4 (1:200, #25229, Cell Signaling Technology), CD11C (1:250, #97585, Cell Signaling Technology), FOXP3 (1:100, #12653, Cell Signaling Technology), murine S100A9 (1:1000, #73425, Cell Signaling Technology), FLT4 (1:250, #AF743, R&D Systems), and phospho-Stat3 (S727) (1:100, #9134, Cell Signaling Technology), followed by incubation with the corresponding biotinylated secondary antibodies (1:250, Vector Laboratories). The ABC kit and DAB kit (Vector Laboratories) were used for detection following the manufacturer's instructions. Sections were

subsequently counterstained with hematoxylin, dehydrated, and mounted using Cytoseal XYL (Richard-Allan Scientific) for microscopy and immunohistochemical analysis.

For automated immunostaining for CD31, pFLT1, and FLT1, paraffin-embedded tissue sections were sectioned at 5 µm and heated at 58 °C for 1 h. Samples were loaded into Leica Bond RX, and sections were dewaxed at 72 °C before being pretreated with EDTA-based epitope retrieval ER2 solution (Leica, AR9640) for 20 min at 100 °C. The rabbit polyclonal antibodies against CD31 (0.08 ug/ml, Abcam, ab182981), pFLT1 (1:50, Millipore, 07-758), or FLT1 (2.5 ug/ml, Invitrogen, MA5-32045) were incubated for 60 min. Samples were then incubated with Leica Bond Post-Primary reagent (rabbit anti-mouse linker, included in the Polymer Refine Detection Kit (Leica, DS9800)) for 8 min, followed by incubation with Leica Bond Polymer (anti-rabbit HRP, included in Polymer Refine Detection Kit (Leica, DS9800)) for another 8 min. Mixed DAB reagent (Polymer Refine Detection Kit) was then incubated for 10 min, and hematoxylin (Refine Detection Kit) counterstaining was performed for 10 min. After staining, sample slides were washed in water, dehydrated using an ethanol gradient (70, 90, and 100%), washed three times in HistoClear II (National Diagnostics, HS-202), and mounted in Permount (Fisher Scientific, SP15). Immunostaining analysis for human samples was performed on sections of paraffin-embedded tissues, which included biopsies or resected samples. Staining was performed with antibodies against human pFLT1 (1:50, Millipore, 07-758) or FLT1 (2.5 ug/ml, Invitrogen, MA5-32045).

For calculating the staining intensity or the number of positively stained cells in tumor sections from mice, QuPath 0.3.2 software (https://qupath.github.io/) was used as previously described (Biswas et al, 2022). The image type was set as Brightfield (H-DAB) to count positive (pos.) cells, and the cell detection channel was set at Hematoxylin + DAB. The DAB threshold was adjusted and optimized for each antibody staining within the fast cell counts feature. To measure different staining intensities, tumor sections were selected and within the positive cell detection feature, detection image was set as optical density sum. Cell: DAB OD mean was used for scoring each compartment. Each threshold was adjusted on a batch-to-batch basis according to the staining condition to minimize false positive/negative readings. The data were normalized to the control group, thus setting the control to 1 in each case. The values for each of the experimental groups are compared relative to their respective control groups. The magnitude of the scale used in the figures is reflective of the relative expression of the experimental group over the control expression.

To assess the expression levels of pFLT1 and total FLT1 in tumor sections from patients, immunostainings were scored by a pathologist (HH) who was blinded to the sample details. The entire field of the tumor section was quantified for each sample. Staining that was scored as 0 or 1 was considered "low" and staining that was scored above 1 (and up to 3) was considered "high", following our previous studies (Acharyya et al, 2012; Biswas et al, 2022).

### Cell culture and in vitro assays

The *Brca1*-def and *Bard1*-def parental tumor cells and derivatives used in this study were cultured in DMEM media supplemented with 10%

FBS and grown at 37 °C in a humidified CO₂ incubator (5% CO₂). All media were supplemented with 100 IU/mL penicillin and 100 μg/mL streptomycin (Life Technologies). They were authenticated by PCR analysis and tested for mycoplasma contamination.

The in vitro viability of sensitive vs. resistant *Brca1-def and Bard1*-def tumor cells with drug treatment was determined by cell viability assay using Promega CellTiter 96® AQueous One Solution Cell Proliferation Assay kit (Promega G3581) following the manufacturer's instructions. Briefly, 1000 *Brca1-def* sensitive or resistant tumor cells, or 500 *Bard1*-def sensitive or resistant tumor cells were plated into each well of a 96-well plate and allowed to grow in growth medium (DMEM supplemented with 10% FBS and Pen-Strep) overnight at 37 °C in a humidified, 5% CO₂ incubator (Keung et al, 2020). Subsequently, cells were treated with either DMSO (vehicle) or various concentrations of talazoparib (prepared in growth medium) ranging from 0 to 10,000 nM for 7 days, replacing with fresh drug-containing media after every 3 days. After 7 days, the drug-containing medium was replaced by adding 100 μl of phenol red-free growth medium and 20 μl of CellTiter 96 AQueous One Solution Reagent (Promega) to each well. After incubation for 1–2 h at 37 °C in a humidified, 5% CO₂ incubator, the absorbance of colored-formazan formation (by metabolically active cells) was measured at 490 nm using a plate reader. The viability was calculated as a percent of absorbance (viable cells) in vehicle-treated controls (designated as 100% viability).

For treatment of cells with recombinant PGF, talazoparib-resistant *Brca1-def* and *Bard1-def Flt1*-proficient and their *Flt1*-deficient tumor cells were plated and cultured overnight in growth medium (DMEM medium supplemented with 10% FBS). For the *Brca1*-def group, cells were then washed three times with HBSS buffer, and serum-free media was added to the plate for 4 h. Cells were subsequently washed once with HBSS and then treated with 50 ng/ml of mouse recombinant PGF (R&D Systems, #465-PL-010) in serum-free medium for 15 s. Cells were then collected, lysed, and prepared for immunoblot analysis. For *Bard1-def* talazoparib-resistant and its *Flt1*i-deficient derivatives, cells were plated and, after an overnight culturing in 10% FBS DMEM medium, washed three times with HBSS buffer and replenished with serum-free medium for 24 h. Cells were then treated with 50 ng/ml mouse recombinant PGF in a serum-free medium for 10 min. Cells were thereafter harvested, and lysates were prepared for immunoblot analysis described below.

## Immunoblot analysis

Cells were washed with cold PBS, collected in lysis buffer consisting of 25 mM Tris, pH 7.5, 150 mM NaCl, 1% Triton X-100 (v/v), 0.5% SDS (w/v), and supplemented with protease inhibitor cocktail (Roche) and phosphatase inhibitor cocktail (Thermo Scientific). Subsequently, the cell suspension was sonicated, and supernatant fraction of cell lysates was collected by centrifugation at $18,000 \times g$ for 10 min at 4 °C. Protein concentration in the supernatant was determined by the BCA protein assay kit (Pierce), and protein sample was prepared by mixing with Laemmli SDS-PAGE reducing sample buffer and incubated at 98 °C for 5 min. After cooling down to room temperature, an equal amount of total protein from each sample was resolved on 4–20% precast polyacrylamide gel (Bio-Rad Cat. # 4561093) by electrophoresis. Protein bands were transferred

to nitrocellulose membranes and blocked with 5% milk in TBST (Tris-buffered saline containing 0.1% Tween-20) by incubating for 1 h at room temperature with constant agitation. Blots were then incubated overnight with primary antibodies (diluted using 2.5% milk in TBST) against pAKT (S473) (1:1000, Cell Signaling Technology, #4060), pSTAT3 (Ser 727) (1:1000, Cell Signaling Technology, #9134) generated in rabbit, or with a mouse mAb against β-actin (1:2000, Sigma, A1978). The membranes were washed three times with TBST, 5 min each time, and incubated for 1 h at room temperature with the corresponding secondary antibodies conjugated with horseradish peroxidase (HRP). The membranes were then washed three times with TBST for 5 min each time before they were developed using an enhanced chemiluminescence (ECL) substrate (Bio-Rad), and specific protein bands were visualized on a Bio-Rad ChemiDoc Imaging System (Bio-Rad). Blotted membranes were stripped with stripping buffer (Thermo Scientific, #46430) for 15 min and subsequently blocked for 1 h with 5% milk in TBST. Membranes were then incubated overnight at 4 °C with antibodies against total AKT (1:1000, Cell Signaling Technology, #4691) and total STAT3 (1:1000, Cell Signaling Technology, #4904), followed by the corresponding horseradish peroxidase (HRP)–conjugated secondary antibodies. The membranes were developed using an ECL substrate (Bio-Rad), and protein bands were visualized on a Bio-Rad ChemiDoc Imaging System (Bio-Rad).

## RNA isolation and qRT-PCR

Total RNA was isolated using TRIzol and RNeasy Mini Kit as previously described (Biswas et al, 2022). RNA (500 ng) was then reverse-transcribed to cDNA using a cDNA Synthesis Kit (Applied Biosystems; Thermo Fisher Scientific). qRT-PCR was performed with 10 ng of cDNA per sample using gene-specific primers and SYBR Green PCR master mix (Applied Biosystems; Thermo Fisher Scientific). GAPDH primers were used as an internal control. qPCR was run using Applied Biosystems 7500 Real-Time PCR system (Applied Biosystems; Thermo Fisher Scientific), and data were exported to Excel (Microsoft) for gene expression analysis using the $2^{-\Delta\Delta Ct}$ method. The qRT-PCR primer sequences used in this study are shown below:

m*Flt1*
forward primer: 5-TGGCTCTACGACCTTAGACTG-3
reverse primer: 5-CAGGTTTGACTTGTCTGAGGTT-3
m*Gapdh*
forward primer: 5-AGGTCGGTGTGAACGGATTTG-3
reverse primer: 5-TGTAGACCATGTAGTTGAGGTCA-3

## Gene repression by CRISPR

The expression of *Flt1* was suppressed by CRISPR/dCas9–KRAB-mediated gene editing following a previously described method (Biswas et al, 2022). We designed the gRNA1 sequence (5'-CAGCGCGTAAGGCAAGACCG-3') and gRNA2 sequence (5'-CAC-CACTAGCACTACCTCCC-3') using the CRISPR-ERA online tool (http://crispr-era.stanford.edu). The forward and reverse oligos were designed based on the gRNA sequence and were then annealed and cloned into the BsmBI-digested LentiCRISPRv2-SFFV-KRAB-dCas9 (Biswas et al, 2022) following the procedure outlined by Feng Zhang's

group (Sanjana et al, 2014). We confirmed the positive clones by PCR using the human U6 forward primer and the reverse oligo of the corresponding gRNA sequence. We produced lentivirus by transfecting the gRNA-cloned lentiviral vector into the Lenti-X 293 T cells line (Takara, cat # 632180) using the third-generation packaging system. Target cells were transduced with viral supernatant (after passing through a 0.45-micron syringe filter) and selected at 48 h post-transduction with puromycin at a final concentration of 8 μg/ml. The efficiency of knockdown was tested by RT-PCR using mouse-*Flt1*-specific primers.

## Expression of *Flt1*

To re-express the *Flt1* gene in *Flt1*-suppressed (CRISPRi) cells, we cloned the cDNA encoding mouse-Flt1 into the lentiviral plasmid pLV-EF1a-IRES-Blast (Addgene #85133; ref. Hayer et al) within BamHI/EcoRI sites. Subsequently, we produced lentiviral particles with the construct using a third-generation packaging system, following standard procedure. To stably express Flt1 in Flt1-repressed cells, we transduced Flt1-repressed cells with lentiviral particles carrying *Flt1* cDNA and selected with blasticidine S hydrochloride antibiotic after 48 h of viral transduction. After 1 week of antibiotic selection, we confirmed the expression of *Flt1* at RNA and protein levels by real-time PCR and western blotting, respectively.

## Patient samples

Human tumor tissue samples from 10 breast cancer patients were collected prior to PARP inhibitor treatment either at the Memorial Sloan Kettering Cancer Center or Emory University School of Medicine in accordance with approved protocols from their respective institutional review boards (IRB), ensuring the protection of patient privacy and conformed to the principles set out in the WMA Declaration of Helsinki and the Department of Health and Human Services Belmont Report. Informed consent was obtained from all subjects participating in this study. The research was conducted following ethical regulations specified by the IRB, and de-identified information was provided to the research team. Tissue sections were used for immunostaining analysis. Appendix Table S1 provides de-identified information about the mutations of tumors from breast cancer patients and indicates whether the tumors had high or low levels of pFLT1. Sections were stained with antibodies against either pFLT1 or total FLT1 and scored as described in the Immunostaining Analysis section after pathological review.

## Statistical analysis

Statistical significance was determined using Prism 9 software (GraphPad Software) to perform the following analyses: (1) unpaired, two-tailed Student's $t$-test, (2) unpaired Welch's $t$-test, (3) unpaired Mann–Whitney $t$-test, (4) one-way ANOVA with post-hoc Tukey's test, and (5) log-rank test. All values were determined as the mean ± SEM, and $P$ values <0.05 were considered statistically significant. Experimental results were obtained after repeating them three times for reproducibility and rigor. Power analysis was performed for sample size calculation for animal experiments. The Kaplan–Meier method was utilized to generate progression-free survival (PFS) curves. To assess the null hypothesis suggesting no difference between the two curves, the log-rank test was employed through the survdiff() function in R.

### The paper explained

**Problems**

Resistance to PARP inhibitors (PARPi) remains a major treatment challenge in *BRCA*1/2-mutant breast cancer. Although several resistance mechanisms involving the DNA damage response (DDR) pathway have been identified, none have been successfully targeted in the clinic.

**Results**

In this study, we generated new in-vivo treatment models of *Brca1*- and *Bard1*-mutant breast cancer that recapitulate the striking response to PARPi, acquired resistance, and recurrence that is observed in patients with breast cancer. Distinct from previously defined resistance mechanisms, we identified a novel PGF-FLT1-AKT signaling pathway of adaptive resistance to PARPi using these models. Mechanistically, activated FLT1 in PARPi-resistant cancer cells promotes cell survival through AKT activation and reduced infiltration of CD8[+] T cells in tumors. Importantly, genetic and pharmacological inhibition of FLT1 using axitinib re-sensitizes resistant cancer cells to PARPi. Consistent with our preclinical studies, FLT1 activation in tumor cells at pre-treatment significantly correlates with shorter progression-free survival on PARPi in patients with breast cancer.

**Impact**

This study, for the first time, identified a potentially actionable PGF-FLT1-AKT axis that mediates PARPi resistance in breast cancer. Of significance, FLT1 expression in cancer cells can be used as a biomarker to stratify patients who might benefit from combination treatment with axitinib and PARPi. Since axitinib is already FDA-approved for renal cancer treatment, it can be re-purposed to enhance the efficacy and durability of PARPi treatment in breast cancer.

## Graphics

Graphics from the schematics in the figures (Figs. 1A,D, 2K, 3C, 4A,D, 7A, and EV3J) and the synopsis were designed using Biorender.com with an institutional license.

## Data availability

This study includes no data deposited in external repositories.

The source data of this paper are collected in the following database record: biostudies:S-SCDT-10_1038-S44321-024-00094-2.

## Peer review information

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

## Acknowledgements

We thank Jedd D Wolchok (Cornell Medicine), Neil Vasan, Shan Zha, Laura Pasqualucci, Alberto Ciccia (CUIMC), and Thordur Oskarsson (Moffitt Cancer Center, Florida) for their feedback in our studies. We thank Jennifer M Robertson from Emory University; Dajiang (Kevin) Sun, Sean Kim, Rohaan Hegde, Timothy Zhong, and Matthias Szabolcs from CUIMC, and members of the Molecular Cytology Facility at MSKCC for their assistance in our studies. This work was supported by the Department of Defense Breast Cancer Breakthrough Award (S Acharyya) and the Irma T Hirschl Monique Weill-Caulier Trust Award (S Acharyya). S Acharyya was supported by the NCI R01 CA231239, American Cancer Society Scholar Award, Susan G Komen Career Catalyst Award, Pershing Square Sohn Award, METAvivor Early Career Investigator Award, Phi Beta Psi Sorority Cancer Research Award, Columbia University Irving Scholars Program, HICCC and Irving Institute Pilot Awards funded through HICCC P30CA013696. These studies utilized the resources from the Herbert Irving Comprehensive Cancer Center (HICCC) core facilities, which are funded in part through Center Grant P30CA013696.

## Author contributions

**Yifan Tai**: Resources; Data curation; Formal analysis; Validation; Investigation; Visualization; Methodology; Writing—original draft; Writing—review and editing. **Angela Chow**: Resources; Data curation; Validation; Investigation; Methodology; Writing—review and editing. **Seoyoung Han**: Data curation; Investigation; Methodology; Writing—review and editing. **Courtney Coker**: Resources; Data curation; Validation; Investigation; Methodology; Writing—review and editing. **Wanchao Ma**: Resources; Data curation; Validation; Investigation; Visualization; Methodology; Writing—review and editing. **Yifan Gu**: Investigation; Methodology; Writing—review and editing. **Valeria Estrada Navarro**: Resources; Data curation; Methodology; Writing—review and editing. **Manoj Kandpal**: Resources; Data curation; Formal analysis; Validation; Investigation; Methodology; Writing—review and editing. **Hanina Hibshoosh**: Resources; Formal analysis; Supervision; Validation; Investigation; Methodology; Writing—review and editing. **Kevin Kalinsky**: Resources; Data curation; Investigation; Writing—review and editing. **Katia Manova-Todorova**: Resources; Data curation; Formal analysis; Validation; Investigation; Writing—review and editing. **Anton Safonov**: Resources; Data curation; Formal analysis; Validation; Methodology; Writing—review and editing. **Elaine M Walsh**: Resources; Data curation; Validation; Investigation; Methodology; Writing—review and editing. **Mark Robson**: Resources; Data curation; Validation; Investigation; Methodology; Writing—review and editing. **Larry Norton**: Resources; Validation; Writing—review and editing. **Richard Baer**: Resources; Data curation; Validation; Visualization; Writing—original draft; Writing—review and editing. **Taha Merghoub**: Resources; Investigation; Methodology; Writing—review and editing. **Anup K Biswas**: Conceptualization; Resources; Formal analysis; Supervision; Validation; Investigation; Visualization; Methodology; Writing—original draft; Writing—review and editing. **Swarnali Acharyya**: Conceptualization; Supervision; Funding acquisition; Investigation; Writing—original draft; Project administration; Writing—review and editing.

Source data underlying figure panels in this paper may have individual authorship assigned. Where available, figure panel/source data authorship is listed in the following database record: biostudies:S-SCDT-10_1038-S44321-024-00094-2.

## Disclosure and competing interests statement

The authors declare no competing interests.

# Expanded View Figures

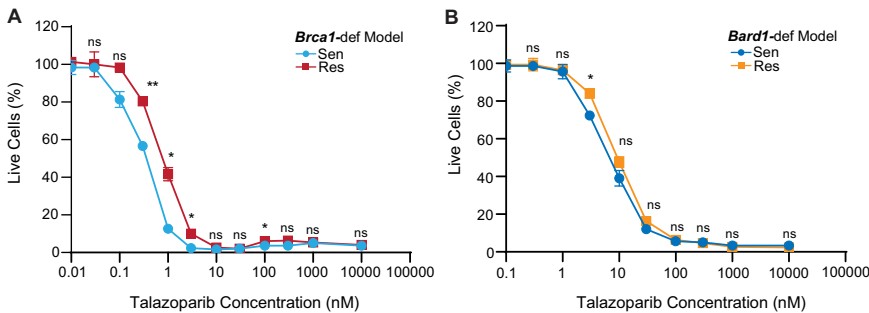

**Figure EV1.    Modest differences in talazoparib sensitivity observed in vitro compared to in-vivo settings (Fig. 1) in *Brca1*-def and *Bard1*-def breast tumor cells.**

(A, B) In vitro cell viability assay comparing the talazoparib sensitivity between talazoparib-sensitive ("Sen") *Brca1*-def and *Bard1*-def breast tumor cells and their talazoparib-resistant ("Res") derivative lines. Cell viability values are normalized to the DMSO-treated control (considered as 100% live cells) and presented as mean values ± SEM. *P* values were determined by a two-tailed, unpaired, Welch's test. For the *Brca1*-def model, $n = 3$ for each concentration tested, ** at 0.3 nM indicates $P = 0.0028$, * at 1 nM indicates $P = 0.0104$, * at 3 nM indicates $P = 0.0104$, and * at 100 nM indicates $P = 0.0357$. For the *Bard1*-def model, $n = 3$ for each tested concentration, and * at 3.0 nM indicates $P = 0.0256$. ns; not significant.

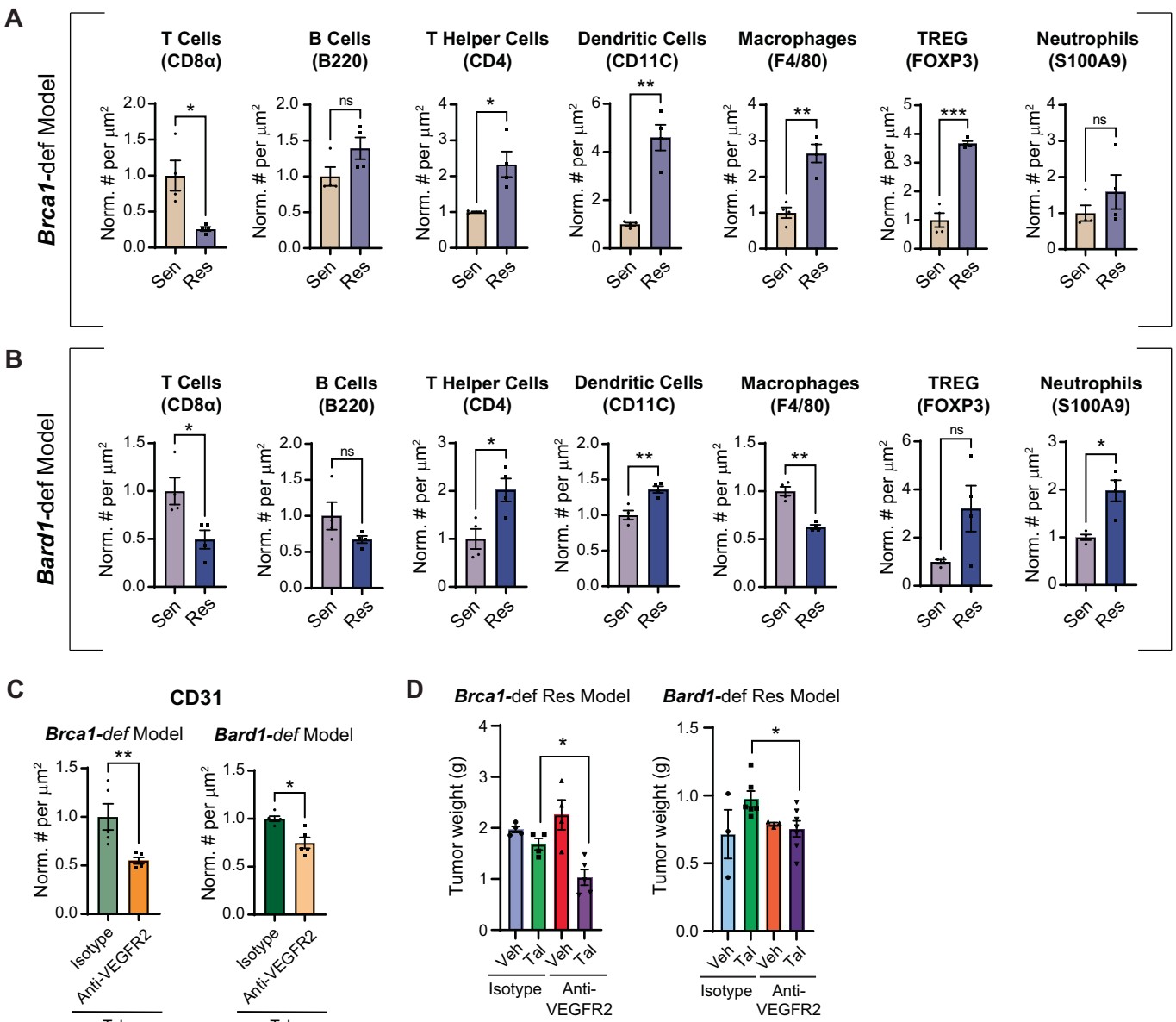

**Figure EV2. Quantification of angiogenesis and immune-cell composition in *Brca1*-def and *Bard1*-def mutant breast tumors by immunostaining analysis.**

(A, B) Immunostained tumor sections from Fig. 1 were quantified using automated QuPath software to identify positively stained cells. Data were presented as mean values ± SEM. *P* values were determined by a two-tailed, unpaired, Welch's test. *n* = 4 mice for talazoparib-sensitive ('Sen") and -resistant ("Res") groups from both models. *P* values are listed in parentheses following the protein analyzed. For the *Brca1*-def model in (A): CD8α (0.0375), CD4 (0.0329), CD11C (0.0061), F4/80 (0.0027), and FOXP3 (0.0007). For the *Bard1*-def model in (B), CD8α (0.0293), CD4 (0.0186), CD11C (0.0048), F4/80 (0.0016), FOXP3 (ns), and S100A9 (0.0182). ns; not significant. (C) Quantitation of immunostained Res tumor sections using antibodies against CD31 (see treatments in Fig. 2I–K) using automated QuPath software to identify positively stained cells. *n* = 5 mice from both groups from both models. Data were presented as mean values ± SEM. *P* values were determined by a two-tailed, unpaired, Mann–Whitney test. ** indicates *P* = 0.0079 for the *Brca1*-def model, and * indicates 0.0159 for the *Bard1*-def model. (D) Tumor weights from Fig. 2J were plotted after collection at endpoint. For the *Brca1*-def model, *n* = 4 tumors for the Isotype + Veh group, *n* = 4 tumors for the Isotype + Tal group, *n* = 4 tumors for the Anti-VEGFR2 + Veh group, and *n* = 5 tumors for the anti-VEGFR2 + Tal group. For the *Bard1*-def model, *n* = 3 tumors for Isotype + Veh group, *n* = 6 tumors for Isotype + Tal group, *n* = 3 tumors for anti-VEGFR2 + Veh group, and *n* = 7 tumors for anti-VEGFR2 + Tal group. Data were presented as mean values ± SEM. *P* values were determined by a two-tailed, unpaired, Student's *t*-test, comparing endpoint tumor weights between the Isotype + Tal and anti-VEGFR2 + Tal groups. * indicates *P* = 0.0141 for the *Brca1*-def model, and * indicates *P* = 0.0222 for the *Bard1*-def model.

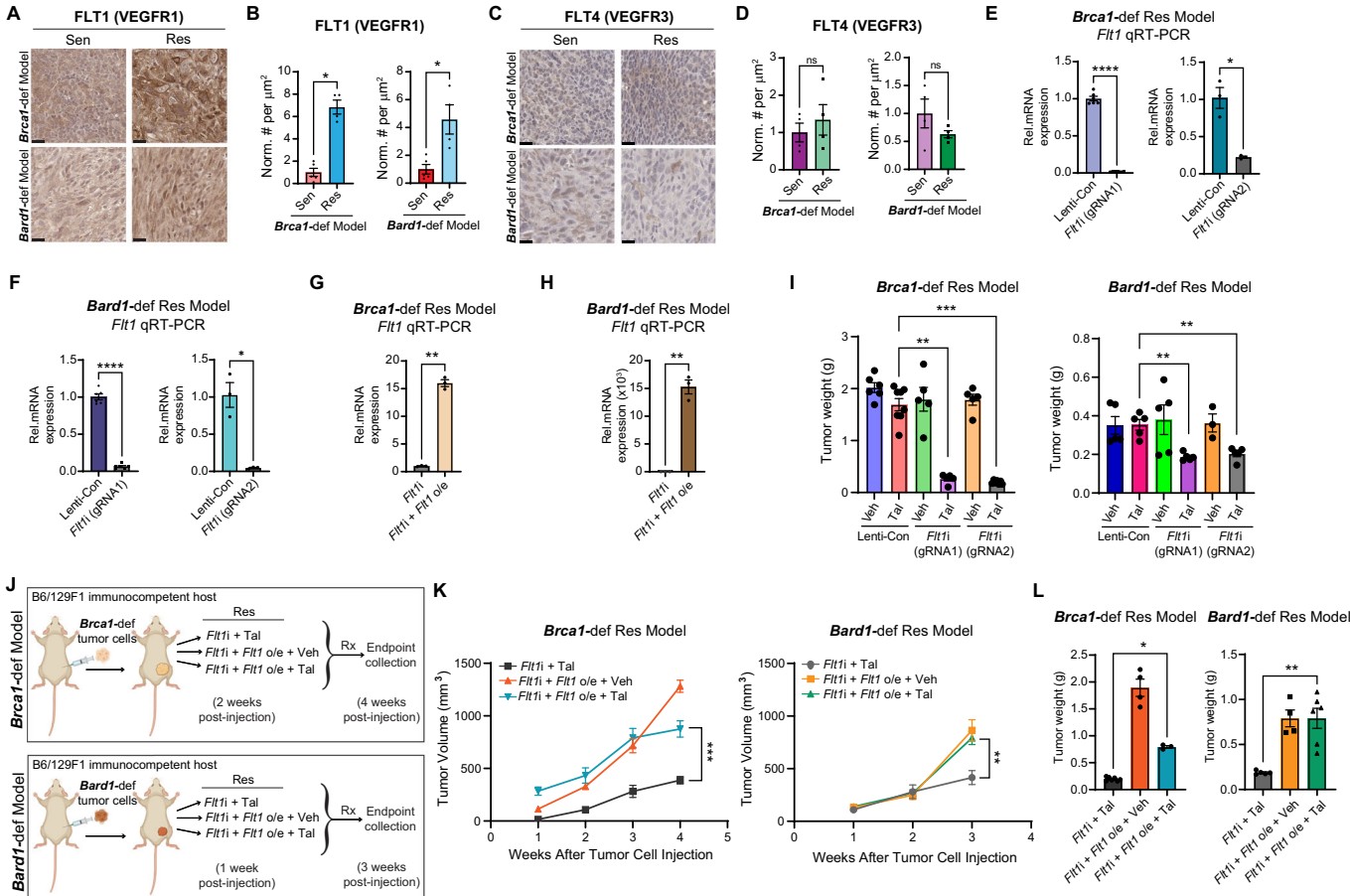

◀ **Figure EV3.  FLT1 promotes PARPi-resistance in the *Brca1*-def and *Bard1*-def breast cancer models.**

(A) Representative images of IHC for total FLT1 expression in tumor sections from the mice described in Fig. 1. Scale bars, 20 μm. (B) Immunostained sections from (A) were quantified using automated QuPath software to identify positively stained cells. $n = 5$ talazoparib-sensitive ("Sen") tumors, and $n = 4$ talazoparib-resistant ("Res") tumors from both *Brca1*-def and *Bard1*-def models. Data were presented as mean values ± SEM. *P* values were determined by a two-tailed, unpaired, Mann–Whitney test. * indicates $P = 0.0159$ for both models. (C), Representative images of IHC for FLT4 in tumor sections from Fig. 1. Scale bars, 20 μm. (D) Immunostained sections from (C) were quantified using automated QuPath software to identify positively stained cells. $n = 4$ Sen and Res tumors for both *Brca1*-def and *Bard1*-def models. Data were presented as mean values ± SEM. *P* values were determined by a two-tailed, unpaired, Welch's test. ns; not significant. (E) qRT-PCR results of *Flt1* repression of the indicated groups in the *Brca1*-def model for both gRNAs. $n = 6$ (consisting of two independent experiments for each triplicate testing) for both Lenti-Con and *Flt1*i for gRNA1 and $n = 3$ (one triplicate testing) for both Lenti-Con and *Flt1*i for gRNA2. Data were presented as mean values ± SEM. *P* values were determined by a two-tailed, unpaired, Welch's test: **** indicates $P < 0.0001$ for gRNA1 and * indicates 0.0285 for gRNA2. (F) qRT-PCR results of *Flt1* repression of the indicated groups in the *Bard1*-def model. $n = 6$ (two independent triplicate testing) for both Lenti-Con and *Flt1*i for gRNA1 and $n = 3$ (one triplicate testing) for both Lenti-Con and *Flt1*i for gRNA2. Data were presented as mean values ± SEM. *P* values were determined by a two-tailed, unpaired, Welch's test: **** indicates $P < 0.0001$ for gRNA1 and * indicates $P = 0.0273$ for gRNA2. (G) qRT-PCR results of *Flt1* expression of the indicated groups in the *Brca1*-def model. $n = 3$ (one triplicate testing) for *Flt1*i and *Flt1*i + *Flt1* overexpression ("o/e") groups. Data were presented as mean values ± SEM. *P* values were determined by a two-tailed, unpaired, Welch's test: ** indicates $P = 0.0016$. (H) qRT-PCR results of *Flt1* expression of the indicated groups in the *Bard1*-def model. $n = 3$ (one triplicate testing) for *Flt1*i and *Flt1*i + *Flt1* o/e groups. Data were presented as mean values ± SEM. *P* values were determined by a two-tailed, unpaired, Welch's test: ** indicates $P = 0.0067$. (I) Tumor weights from Fig. 3D were plotted at endpoint. For the *Brca1*-def model, $n = 6$ tumors for Lenti-Con + Veh, $n = 8$ tumors for Lenti-Con + Tal, $n = 5$ tumors for *Flt1*i (gRNA1) + Veh or Tal and *Flt1*i (gRNA2) + Veh, and $n = 7$ tumors for *Flt1*i (gRNA2) + Tal treatment groups. For the *Bard1*-def model, $n = 5$ tumors for Lenti-Con + Veh or Tal, *Flt1*i (gRNA1) + Veh or Tal, and *Flt1*i (gRNA2) + Tal, and $n = 3$ tumors for *Flt1*i (gRNA2) + Veh. Data were presented as mean values ± SEM. *P* values were determined by a two-tailed, unpaired, Mann–Whitney test, comparing endpoint tumor weights between Lenti-Con + Tal and *Flt1*i (gRNA1 or gRNA2) + Tal groups. For the *Brca1*-def model, ** indicates $P = 0.0016$ between Lenti-Con + Tal and *Flt1*i (gRNA1) + Tal and *** indicates $P = 0.0003$ between Lenti-Con + Tal and *Flt1*i (gRNA2) + Tal. For the *Bard1*-def model, ** indicates $P = 0.0079$ between Lenti-Con + Tal and *Flt1*i (gRNA1 or gRNA2) + Tal. (J) Schematic representation of the experiment designed to test whether *Flt1* re-expression rescues talazoparib-resistance in *Brca1*-def and *Bard1*-def mammary tumors with *Flt1* repression. The generation of *Brca1*- and *Bard1*-def cancer cells were described in Fig. 3. To stably re-express *Flt1*, we transduced *Flt1*-repressed cells with lentiviral particles carrying *Flt1* cDNA. For the *Brca1*-def model, randomized mice received either vehicle ("Veh") or talazoparib ("Tal") treatment starting at 2 weeks after tumor-cell injection and were euthanized at 4 weeks following injection. For the *Bard1*-def model, randomized mice received treatment at 1 week after tumor-cell injection and were euthanized at 3 weeks following injection. (K) Tumor growth curves comparing *Flt1*i + Tal and *Flt1*i-*Flt1* o/e + Veh or Tal. For the *Brca1*-def model, $n = 7$ mice for *Flt1*i + Tal, $n = 4$ mice for *Flt1*i-*Flt1* + o/e + Veh, and $n = 3$ mice for *Flt1*i-*Flt1* + o/e + Tal. For the *Bard1*-def model, $n = 5$ mice for *Flt1*i + Tal, $n = 4$ mice for *Flt1*i-*Flt1* + o/e + Veh, and $n = 6$ mice for *Flt1*i-*Flt1* + o/e + Tal. Data were presented as mean values ± SEM. *P* values were determined with a one-way ANOVA test, comparing endpoint tumor volumes between the *Flt1*i + Tal and *Flt1*i-*Flt1* + o/e + Tal groups. For the *Brca1*-def model, *** at 4 weeks indicates $P = 0.0001$ and for the *Bard1*-def model, ** at 3 weeks indicates $P = 0.0074$. (L) Tumor weights from K were plotted at endpoint. For the *Brca1*-def model, $n = 7$ tumors for *Flt1*i + Tal, $n = 4$ tumors for *Flt1*i-*Flt1* + o/e + Veh, and $n = 3$ tumors for *Flt1*i-*Flt1* + o/e + Tal. For the *Bard1*-def model, $n = 5$ tumors for *Flt1*i + Tal, $n = 4$ tumors for *Flt1*i-*Flt1* + o/e + Veh, and $n = 6$ tumors for *Flt1*i-*Flt1* + o/e + Tal. Data were presented as mean values ± SEM. *P* values were determined by a two-tailed, unpaired, Mann–Whitney test, comparing endpoint tumor weights between the *Flt1*i + Tal and *Flt1*i-*Flt1* + o/e + Tal groups. For the *Brca1*-def model, * indicates $P = 0.0167$ and for the *Bard1*-def model, ** indicates $P = 0.0043$.

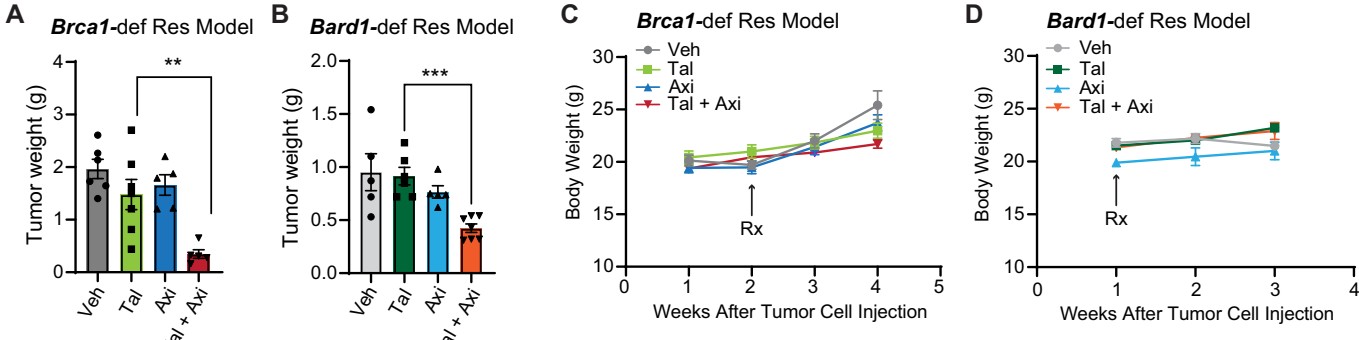

**Figure EV4. Tumor and body weight analysis of mice treated with vehicle ("Veh"), talazoparib ("Tal"), and/or axitinib ("Axi").**

(A) Tumor weights from Fig. 4B were plotted at endpoint. $n = 6$ Veh-treated tumors, $n = 7$ Tal-treated tumors, $n = 5$ Axi-treated tumors, and $n = 5$ tumors treated with Tal + Axi. Data were presented as mean values ± SEM. $P$ values were determined by a two-tailed, unpaired, Mann–Whitney test, comparing endpoint tumor weights between the Tal and Tal + Axi groups. ** indicates $P = 0.0051$. (B) Tumor weights from Fig. 4E were plotted after collection at endpoint. $n = 5$ Veh-treated tumors, $n = 6$ Tal-treated tumors, $n = 5$ Axi-treated tumors, and $n = 7$ tumors treated with Tal + Axi. Data were presented as mean values ± SEM. $P$ values were determined by a two-tailed, unpaired Mann–Whitney test comparing endpoint tumor weights between the Tal and Tal + Axi groups. *** indicates $P = 0.0006$. (C, D) Body weight from treatment initiation until endpoint from Fig. 4A, D for each experiment. For the *Brca1*-def model, $n = 6$ Veh-treated tumors, $n = 7$ Tal-treated tumors, $n = 5$ Axi-treated tumors, and $n = 5$ tumors treated with Tal + Axi. For the *Bard1*-def model, $n = 5$ Veh-treated tumors, $n = 6$ Tal-treated tumors, $n = 5$ Axi-treated tumors, and $n = 7$ tumors treated with Tal + Axi.

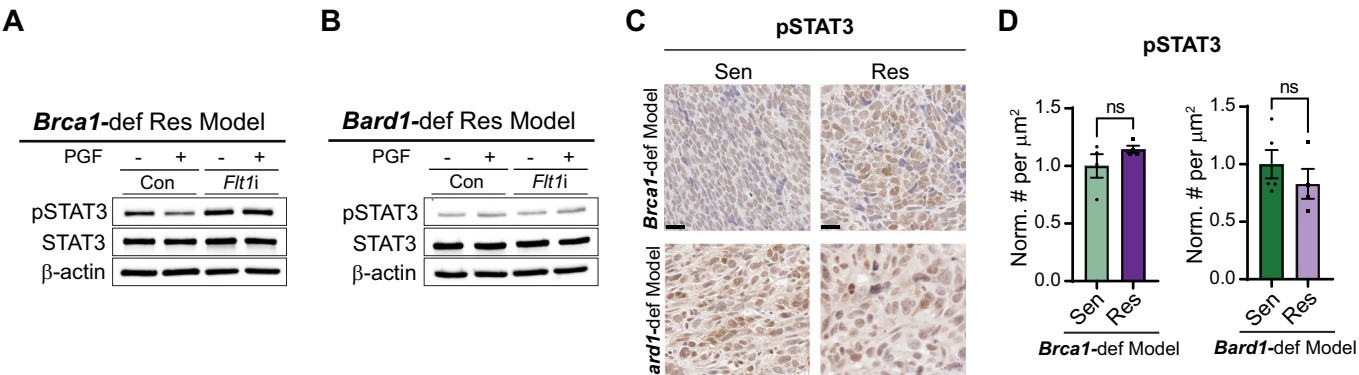

**Figure EV5. Quantitation of phosphorylated STAT3 (pSTAT3) levels in *Brca*1-def and *Bard*1-def breast tumor cells and tumor tissue sections.**

(A, B) Immunoblot analysis was performed using antibodies against pSTAT3, STAT3 and β-actin (loading control) using lysates from *Flt1*-expressing (Con) and -deficient (*Flt1*i), talazoparib-resistant ("Res") *Brca1* and *Bard1*-def tumor cells, treated with 50 ng/mL of mouse PGF protein that were used in Fig. 5A, B. (C) Representative images of IHC for pSTAT3 staining in tumor sections from the mice described in Fig. 1 comparing talazoparib-sensitive ("Sen") tumors to talazoparib-resistant ("Res") tumors. Scale bars, 20 μm. (D) Immunostained sections from (C) were quantified using automated QuPath software to identify positively stained cells. For the *Brca1*-def model, n = 4 tumors for both Sen and Res. For the *Bard1*-def model, n = 5 Sen tumors, and n = 4 Res tumors. Data were presented as mean values ± SEM. P values were determined by a two-tailed, unpaired, Mann–Whitney test. ns; not significant.

