## [Peer Review File · EMBO Molecular Medicine]

FLT1 activation in cancer cells promotes PARP-inhibitor resistance in breast cancer

Swarnali Acharyya, Anup Biswas, Yifan Tai, Angela Chow, Seoyoung Han, Courtney Coker, Wanchao Ma, Yifan Gu, Valeria Estrada Navarro, Manoj Kandpal, Hanina Hibshoosh, Kevin Kalinsky, Katia Manova-Todorova, Anton Safonov, Elaine Walsh, Mark Robson, Larry Norton, Richard Baer, and Taha Merghoub

Corresponding author(s): Swarnali Acharyya (sa3141@cumc.columbia.edu) , Anup Biswas (akb2180@cumc.columbia.edu)

Review Timeline:

Submission Date:	6th Sep 23
Editorial Decision:	5th Oct 23
Revision Received:	3rd May 24
Editorial Decision:	29th May 24
Revision Received:	5th Jun 24
Accepted:	7th Jun 24

Editor: Lise Roth

Transaction Report:

5th Oct 2023

Dear Dr. Acharyya,

Thank you for the submission of your manuscript to EMBO Molecular Medicine. We have now received feedback from the three reviewers who agreed to evaluate your manuscript. As you will see below, the reviewers raise substantial concerns on your work, which unfortunately preclude its publication in EMM in its current form.

The reviewers find that the question addressed by the study is of potential interest, however they remain unconvinced that some of the major conclusions are sufficiently supported by the data. They thus raise issues related (but not limited) to the reliance on loss-of-function experiments, lack of control/rescue experiment regarding Flt1 suppression, insufficient mechanistic insight, lack of comparison with other inhibitors, and unclear human data.

After further consultation with the referees, they agreed that the phospho-proteomic analyses and organoid experiments could be optional for revisions. Regarding the novelty, we would like you to clearly discuss the translational advance brought forward in your manuscript.

If you feel you can satisfactorily address the referees' points, you may wish to submit a revised version of your manuscript. Please attach a covering letter giving details of the way in which you have handled each of the points raised by the referees. A revised manuscript will once again be subject to review, and we cannot guarantee at this stage that the eventual outcome will be favorable.

We are expecting your revised manuscript within three months, if you anticipate any delay, please contact us.

We require:

- 1) A .docx formatted version of the manuscript text (including legends for main figures, EV figures and tables). Please make sure that the changes are highlighted to be clearly visible.
- 2) Individual production quality figure files as .eps, .tif, .jpg (one file per figure). For guidance, download the 'Figure Guide PDF' (<https://www.embopress.org/page/journal/17574684/authorguide#figureformat>).
- 3) At EMBO Press we ask authors to provide source data for the main figures. Our source data coordinator will contact you to discuss which figure panels we would need source data for and will also provide you with helpful tips on how to upload and organize the files.
- 4) A .docx formatted letter INCLUDING the reviewers' reports and your detailed point-by-point responses to their comments. As part of the EMBO Press transparent editorial process, the point-by-point response is part of the Review Process File (RPF), which will be published alongside your paper.
- 5) A complete author checklist, which you can download from our author guidelines (<https://www.embopress.org/page/journal/17574684/authorguide#submissionofrevisions>). Please insert information in the checklist that is also reflected in the manuscript. The completed author checklist will also be part of the RPF.
- 6) Please note that all corresponding authors are required to supply an ORCID ID for their name upon submission of a revised manuscript. An ORCID identifier is currently missing for Anup K Biswas.
- 7) It is mandatory to include a 'Data Availability' section after the Materials and Methods. Before submitting your revision, primary datasets produced in this study need to be deposited in an appropriate public database, and the accession numbers and database listed under 'Data Availability'. Please remember to provide a reviewer password if the datasets are not yet public (see <https://www.embopress.org/page/journal/17574684/authorguide#dataavailability>).

In case you have no data that requires deposition in a public database, please state so in this section (This study includes no data deposited in external repositories).

Note that the Data Availability Section is restricted to new primary data that are part of this study.

8) For data quantification: please specify the name of the statistical test used to generate error bars and P values, the number (n) of independent experiments (specify technical or biological replicates) underlying each data point and the test used to calculate p-values in each figure legend. The figure legends should contain a basic description of n, P and the test applied. Graphs must include a description of the bars and the error bars (s.d., s.e.m.). Please provide exact p values.

13) Author contributions: CRediT has replaced the traditional author contributions section because it offers a systematic machine readable author contributions format that allows for more effective research assessment. Please remove the Authors Contributions from the manuscript and use the free text boxes beneath each contributing author's name in our system to add specific details on the author's contribution. More information is available in our guide to authors.

16) As part of the EMBO Publications transparent editorial process initiative (see our Editorial at <http://embomolmed.embopress.org/content/2/9/329>), EMBO Molecular Medicine will publish online a Review Process File (RPF) to accompany accepted manuscripts.

In the event of acceptance, this file will be published in conjunction with your paper and will include the anonymous referee reports, your point-by-point response and all pertinent correspondence relating to the manuscript. Let us know whether you agree with the publication of the RPF and as here, if you want to remove or not any figures from it prior to publication. Please note that the Authors checklist will be published at the end of the RPF.

I look forward to receiving your revised manuscript.

Yours sincerely,

Lise Roth

***** Reviewer's comments *****

Referee #1 (Comments on Novelty/Model System for Author):

The VEGFR-PI3K-AKT pathway has been reported to regulate PARPi resistance and CD8+ T cell exhaustion. Moreover, PARPi in combination with VEGFRi (with or without immune checkpoint inhibitors) has been tested in clinical trials. The model systems heavily rely on loss-of-function models.

Referee #1 (Remarks for Author):

In this manuscript, the authors reported that VEGFR1 in breast cancer cells promotes PARPi resistance through AKT pathway activation and CD8+ T cell depletion. I have the following comments:

-The VEGFR-PI3K-AKT pathway has been found to play important roles in PARPi resistance of tumor cells, and inhibitors targeting this pathway combined with PARPi have been evaluated in clinical trials. For example, the combination of olaparib and cediranib, a VEGFR1-3 inhibitor, has been demonstrated to be clinically superior to olaparib monotherapy in recurrent platinum-sensitive ovarian cancer (The Lancet Oncology 2014, PMID: 25218906).

-It has been reported that activation of the VEGFA-VEGFR pathway promotes CD8+ T cell exhaustion in the tumor microenvironment, leading to immunotherapy resistance (e.g., J Exp Med. 2015, PMID: 25601652). The PD-L1 inhibitor, durvalumab, in combination with olaparib and cediranib, has been tested in recurrent women's cancers (J Immunother Cancer 2019, PMID: 31345267). Thus, the novelty of this manuscript seems limited.

-This manuscript mainly relies on loss-of-function experiments. Since the title is "FLT1 activation in tumor cells drives PARP-inhibitor resistance in breast cancer", more data are needed to directly demonstrate this conclusion.

Referee #2 (Comments on Novelty/Model System for Author):

While FLT1/VEFG1 has been implicated in driving tumor growth and/or metastasis in other cancers, this appears to be the first time it has been implicated in adaptive resistance to poly (ADP-ribose) polymerase (PARP) inhibitors. Further, in contrast to previous work showing the role of secondary mutations in PARP inhibitor (PARPi) resistance, this work shows a putative role for the PGF-FLT1-AKT pathway in mediating PARPi resistance that is potentially actionable. So along with the models these authors have developed, novelty and medical impact are deemed high and the model systems are adequate.

Referee #2 (Remarks for Author):

This manuscript by Tai et al. (EMM-2023-18653) addresses the issue of mechanisms, aside of secondary mutations in

homology-dependent repair (HDR) genes, in the development of resistance of breast cancers to poly (ADP-ribose) polymerase (PARP) inhibitors despite an initial response. For this purpose, the authors utilize tumor cells from mice with conditional deletion of Brca1 or Bard1, then do orthotopic transplants into mammary glands and develop PARP inhibitor (PARPi) resistance in vivo via exposure to the PARPi talazoparib. PARPi sensitive controls were not exposed to talazoparib. On this basis, the authors find that PARPi resistance in mice is associated with increased expression of placental growth factor (PGF), its ligand FLT1/VEGF, and activation of a downstream target, AKT. There is some corroboration of these effects in human PARPi resistant breast cancers that have mutations in the HDR genes BRCA1, BRCA2 or PALB2. Additionally, the authors show that genetic or pharmacological inhibition of FLT1 in PARPi resistant tumors restores sensitivity and leads to T-cell dependent regression of tumors in immunocompetent mice in the presence of PARPi. Further, the authors demonstrate that in vitro treatment of PARPi resistant tumor cells from mice increases levels of phospho-FLT1 (pFLT1). While the innovative models are appreciated and elucidation of a PGF-FLT1-AKT signaling axis in PARPi resistant breast cancer appears to be novel, and is medically relevant and has translational implications, there are concerns, detailed below, with certain aspects of this study that should be addressed.

Major issues:

1. To support a role for FLT1 in mediating PARPi resistance, in Fig. 3, Flt1 is suppressed using a single gRNA carried by a lentivirus. There is no consideration of possible off-target effects, much less rescue by re-expressing Flt1 or increasing confidence with use of a 2nd, independent, gRNA.
2. While Figs. 2E-F and 5E-F show that both Brca1-deficient and Bard1-deficient tumors that are resistant to PARPi have increased PGF and pAKT, respectively, as compared to PARPi sensitive tumors, further support to corroborate the existence of a PGF-FLT1-AKT1 pathway in PARPi resistance is lacking. For example, under conditions of suppression of Flt1 in PARPi resistant tumors from mice, is pAKT suppressed? Or, if one compares human breast cancers (such as in Figs. 3G-H and 7B that have high pFLT1), do pAKT levels correlate with levels of pFLT1?
3. Although the extension to human breast cancers in Figs. 3G-H and Fig. 7 is important, there are concerns. For example, Fig. 3G indicates that "tissue sections from PARPi-resistant tumors" were utilized in Fig. 3H. However, Fig. 3H stratifies progression-free survival in these PARPi-resistant tumors by high or low levels of pFLT1. If all of the tumors are PARPi-resistant, maybe differences in pFLT1 (FLT1 activation) are not driving resistance but merely controlling the growth rate of the tumors. Can clarification be added to address this point? Also, while Suppl. Tables 1 and 2 indicate that human breast cancer samples utilized in Figures 3G-H and 7, respectively, had mutations in either BRCA1, BRCA2 or PALB2, the Results section and the figure legends do not appear to indicate which of the high vs low pFLT1 tumors had mutations in each gene. This is particularly important since the only extension of the results in mice for Brca1- and Bard1-deficiency to these other key HDR genes (BRCA1 and PALB2) is in these experiments utilizing human tumors. Additionally, the presentation for these human tumors would be strengthened by showing examples (images) of tumors with high vs low pFLT1.

Minor issues:

1. Why, as shown in Figs. 1B&E do Bard1-deficient tumors grow more rapidly than Brca1-deficient tumors in the absence of PARPi?
2. In many figures, such as in Figs. 2B, 2D, 2F and 3B the scales are different, sometimes dramatically so, when comparing Brca1-deficient vs Bard1-deficient cells/tumors in the same figure part. At a minimum, the authors should acknowledge this and give some justification for presenting the data in this way.
3. While both Brca1-deficient and Bard1-deficient tumors show clear differences between the growth/survival of resistant vs sensitive cells in vivo, they argue the differences are less pronounced in vitro (Suppl. Fig. 1). In actuality, in vitro, it looks as though there are clear and significant differences for Brca1-deficient cells but not Bard1-deficient tumor cells. Rather than indicating (as in the title to Suppl. Fig. 1) that both Brca1- and Bard1-deficient cells show "modest differences in talazoparib sensitivity", it might be clearer to acknowledge that these two genotypes do not appear to behave in the same way in vitro.
4. Either the text of the Results section or the legend to figure 2 should clarify how endothelial cells are being identified for this figure, on the basis of being CD31+, by morphology, or by both.
5. Suppl. Figure 2 compares Brca1- and Bard1-deficient tumors that are sensitive or resistant to PARPi for immune cell composition. The authors should acknowledge that for certain types of immune cells, such as B cells and macrophages, that there are not consistent differences between sensitive and resistant tumors in the two distinct genotypes.
6. While Suppl. Fig. 3B quantifies the number of FLT1+ cells in a specific area, based on the examples shown in Suppl. Fig. 3A, it would appear that the mean intensity of FLT1 is much higher in PARPi resistant tumors from a Brca1-deficient genetic background than for Bard1-deficient tumors. Either more representative examples are needed or there some clarification should be added.
7. Neither the text of the Results section nor the legend to Fig. 6 appears to indicate what model of T-cell deficient mice is

utilized.

Referee #3 (Comments on Novelty/Model System for Author):

The experiments reported here are mainly performed using in vivo models. As also suggested in my report, the use of organoids derived from BC patients may be an alternative for some experiments. However, I believe that the authors have done a great job and the results are very interesting and with a great potential.

Referee #3 (Remarks for Author):

The authors identify a novel mechanism underlying the resistance to PARPi in breast cancers, that through the activation of the FLT1 and AKT leads to cell survival of BRCA1/2-mutant breast cancer. The experiments are performed mostly in in vivo models, that is remarkable, and strongly support the conclusions drawn in the paper. The text is well-written and straightforward. The topic they face is of great importance. Therefore, overall, I believe that the work presented is suitable for publication in EMM.

However, I have some comments/suggestions and requests, more specifically from a mechanistic point of view.

The data clearly demonstrate that AKT is activated, conferring pro-survival signals. I suppose that the authors checked the activation of the S473 residue (since this ab has been listed in the "Material" section).

1. Can the authors indicate the specific residue?
2. What about the activation of the T308?
3. Does FLT1 directly activate AKT?

On the same line, and considering that the PI3K pathway is a well-recognized mechanism of BC resistance, can the authors clarify if there is any link between these two pathways?

Alpelisib is a well-known drug used in combination with PARPi.

Did the authors check if alpelisib+PARPi phenocopy the effects given by axitinib+PARPi? And, more importantly, do AKT inhibitors (e.g. capivasertib) phenocopy the effects given by axitinib+PARPi? This would strongly demonstrate that the effects here described are through the FLT1-AKT pathway.

Moreover, are there additional targets of the FLT1 activation that contribute to the effects you describe? Maybe a phospho-proteomic analysis may be helpful.

An additional point I would like to highlight is on the use of the PARP inhibitor. The PARP inhibitor used by the authors is talazoparib, that can block the activity of both PARP1 and PARP2. Recent literature shows that inhibition of PARP2 caused side effects at hematopoietic level; a novel and highly selective PARP1 inhibitor is now available (AZD530). I would suggest to add the use of this inhibitor in key experiments.

To further strengthen their findings, could the author use the proposed therapy in organoids derived from BC patient resistant to PARPi?

Point-by-point response to reviewers (in blue).

Reviewer#1:

The VEGFR-PI3K-AKT pathway has been reported to regulate PARPi resistance and CD8+ T cell exhaustion. Moreover, PARPi in combination with VEGFRi (with or without immune checkpoint inhibitors) has been tested in clinical trials. The model systems heavily rely on loss-of-function models.

Referee #1 (Remarks for Author):

In this manuscript, the authors reported that VEGFR1 in breast cancer cells promotes PARPi resistance through AKT pathway activation and CD8+ T cell depletion. I have the following comments:

-The VEGFR-PI3K-AKT pathway has been found to play important roles in PARPi resistance of tumor cells, and inhibitors targeting this pathway combined with PARPi have been evaluated in clinical trials. For example, the combination of olaparib and cediranib, a VEGFR1-3 inhibitor, has been demonstrated to be clinically superior to olaparib monotherapy in recurrent platinum-sensitive ovarian cancer (The Lancet Oncology 2014, PMID: 25218906).

-It has been reported that activation of the VEGFA-VEGFR pathway promotes CD8+ T cell exhaustion in the tumor microenvironment, leading to immunotherapy resistance (e.g., J Exp Med. 2015, PMID: 25601652). The PD-L1 inhibitor, durvalumab, in combination with olaparib and cediranib, has been tested in recurrent women's cancers (J Immunother Cancer 2019, PMID: 31345267). Thus, the novelty of this manuscript seems limited.

-This manuscript mainly relies on loss-of-function experiments. Since the title is "FLT1 activation in tumor cells drives PARP-inhibitor resistance in breast cancer", more data are needed to directly demonstrate this conclusion.

Authors: We thank the reviewer for the comments. We have now performed additional experiments to address the concerns, and the data has been incorporated into the revised manuscript. The points of similarity and differences with VEGFR-driven ovarian cancer and the novelty of our studies have been clarified in the Discussion. These points are summarized below.

(i) Reliance on loss-of-function experiments and more data needed to support the conclusions: In our revised manuscript, we have incorporated two independent guide RNAs against FLT1 in both mouse models (Brca1- and Bard1-deficient) to further strengthen our conclusions. Our new data (EV3E-F, EV3I and Fig. 3D-E) shows that Flt1 suppression by both gRNAs re-sensitizes PARP inhibitor-resistant tumors to talazoparib treatment in 2 independent models. We further generated FLT1-overexpression constructs to the FLT1-knockdown background for gain-of-function experiments. These experiments showed that FLT1-re-expression in FLT1-suppressed tumors reverses FLT1-suppression-induced tumor regression in the presence of talazoparib in both models (EV3G and 3J-L). Our data support the conclusion that FLT1 is a mediator of PARP-inhibitor resistance in these breast cancer models.

(ii) The known contribution of VEGFR signaling in ovarian cancer, the existing clinical trial data on VEGFR inhibition in ovarian cancer patients and its link to immunotherapy response, which are limiting the novelty of the current study on breast cancer:

We acknowledge this important point raised by the reviewer. The role of VEGFR signaling in tumor growth and during PARP inhibition is indeed well-established in ovarian cancer models and patients, primarily in the context of VEGFR2-driven angiogenesis. In contrast, from our genetic loss- and gain-of-function and VEGFR2-depletion studies, we find that VEGFR1 signaling in cancer cells (rather than VEGFR2-driven angiogenesis) promotes PARP inhibitor resistance in breast cancer. We think this is an important distinction to highlight in our studies, which was written in the Discussion section of our original manuscript and has been further clarified in our revised version. Since pan-VEGFR inhibitors such as cediranib or axitinib do not allow us to differentiate between the contribution of VEGFR1 or VEGFR2, we performed genetic suppression of FLT1 in cancer cells, which allowed

us to identify its novel function in PARP inhibitor resistance in breast cancer for the first time. We compared these results with antibody-mediated depletion of VEGFR2 (abundantly expressed in endothelial cells), which reduced angiogenesis but did not significantly reduce PARP inhibitor resistance in our models. We think this is especially important since FLT1 or VEGFR1 activation in cancer cells is often overlooked in cancer studies. However, this might be clinically relevant since FLT1 activation in cancer cells can be a useful biomarker for screening tumors that might develop PARPi resistance. Our clinical data (Fig. 7) suggests that FLT1 activation in cancer cells could be used to stratify patients who might benefit from VEGFR inhibition. Therefore, the FLT1 pathway could serve both as a biomarker and therapeutic target for overcoming PARPi resistance in breast cancer.

Regarding the existing literature on T-cell exhaustion and immunotherapy resistance with VEGFR, these studies were performed primarily in the context of VEGF-A signaling and do not shed light on the contribution of PGF-FLT1 signaling in cancer cells. In our studies, the genetic blockade of FLT1 in cancer cells allowed us to identify a novel impact on CD8+ T-cell number in breast cancer models. Unlike VEGF-A signaling effects on T-cell exhaustion, we find that blocking PGF-FLT1 signaling in cancer cells increases the CD8+ T-cell numbers in these tumors and promotes tumor regression (Fig. 6A-B and Fig. 6F-G), an effect that is reversed in T-cell-deficient mice (Fig. 6C-E). Our unpublished preliminary data suggest that FLT1 blockade in cancer cells alters the chemokine landscape of tumors, which allows greater recruitment of CD8+ T-cells and induces cytotoxicity in tumors. Thus, depending on the type of VEGF ligands (VEGF-A vs. PGF) and engagement of their respective receptors (VEGFR2 vs. FLT1, respectively), the impact on the immune system can be very different. We have updated the Discussion to clarify these points.

Referee #2 (Comments on Novelty/Model System for Author):
Responses to Reviewer 2 (in blue).

While FLT1/VEGF1 has been implicated in driving tumor growth and/or metastasis in other cancers, this appears to be the first time it has been implicated in adaptive resistance to poly (ADP-ribose) polymerase (PARP) inhibitors. Further, in contrast to previous work showing the role of secondary mutations in PARP inhibitor (PARPi) resistance, this work shows a putative role for the PGF-FLT1-AKT pathway in mediating PARPi resistance that is potentially actionable. So along with the models these authors have developed, novelty and medical impact are deemed high and the model systems are adequate.

Authors: We appreciate the encouraging comments about the novelty of these studies.

Referee #2 (Remarks for Author):

This manuscript by Tai et al. (EMM-2023-18653) addresses the issue of mechanisms, aside of secondary mutations in homology-dependent repair (HDR) genes, in the development of resistance of breast cancers to poly (ADP-ribose) polymerase (PARP) inhibitors despite an initial response. For this purpose, the authors utilize tumor cells from mice with conditional deletion of Brca1 or Bard1, then do orthotopic transplants into mammary glands and develop PARP inhibitor (PARPi) resistance in vivo via exposure to the PARPi talazoparib. PARPi sensitive controls were not exposed to talazoparib. On this basis, the authors find that PARPi resistance in mice is associated with increased expression of placental growth factor (PGF), its ligand FLT1/VEGF, and activation of a downstream target, AKT. There is some corroboration of these effects in human PARPi resistant breast cancers that have mutations in the HDR genes BRCA1, BRCA2 or PALB2. Additionally, the authors show that genetic or pharmacological inhibition of FLT1 in PARPi resistant tumors restores sensitivity and leads to T-cell dependent regression of tumors in immunocompetent mice in the presence of PARPi. Further, the authors demonstrate that in vitro treatment of PARPi resistant tumor cells from mice increases levels of phospho-FLT1 (pFLT1). While the innovative models are appreciated and elucidation of a PGF-FLT1-AKT signaling axis in PARPi resistant breast cancer appears to be novel, and is medically relevant and has translational implications, there are concerns, detailed below, with certain aspects of this study that should be addressed.

Major issues:

1. To support a role for FLT1 in mediating PARPi resistance, in Fig. 3, Flt1 is suppressed using a single gRNA carried by a lentivirus. There is no consideration of possible off-target effects, much less rescue by re-expressing Flt1 or increasing confidence with use of a 2nd, independent, gRNA.

Authors: We appreciate this suggestion by the reviewer and have performed additional experiments to address this concern. Our new data (EV3 E-F, EV3I and Fig. 3D-E) shows that Flt1 suppression by both gRNAs in 2 independent models re-sensitizes PARP inhibitor-resistant tumors to talazoparib treatment. We further generated FLT1-overexpression constructs to the FLT1-knockdown background for gain-of-function experiments in both models. These experiments showed that FLT1-re-expression in FLT1-suppressed tumors reverses FLT1-suppression-induced tumor regression in the presence of talazoparib (EV3G and EV3J-L). Our data support the conclusion that FLT1 is a mediator of PARP-inhibitor resistance in these breast cancer models.

2. While Figs. 2E-F and 5E-F show that both Brca1-deficient and Bard1-deficient tumors that are resistant to PARPi have increased PGF and pAKT, respectively, as compared to PARPi sensitive tumors, further support to corroborate the existence of a PGF-FLT1-AKT1 pathway in PARPi resistance is lacking. For example, under conditions of suppression of Flt1 in PARPi resistant tumors from mice, is pAKT suppressed? Or, if one compares human breast cancers (such as in Figs. 3G-H and 7B that have high pFLT1), do pAKT levels correlate with levels of pFLT1?

Authors: We agree this experiment would be informative. Figs. 2E- F and Figs. 5C-D show increased PGF and pAKT in the PARPi-resistant tumors for both models. We have corroborated these findings in the tumors lacking FLT1 through either genetic (Fig. 5E-F) or pharmacologic (Fig. 5G-H) inhibition of FLT1.

Regarding patient samples from BRCA-deficient patients, these are a rare subset. When pre-treatment biopsies are obtained from these patients, they are prioritized for pathological evaluation, and any residual slides are kept for research. To overcome these limitations, we collaborated with two centers (MSKCC and Emory) to get samples for clinical validation of our findings. We had only two blank sections per patient left over after clinical evaluation, on which we performed phospho- and total-FLT1 immunohistochemistry (Fig. 7 and Appendix Fig. S2). Therefore, we were unable to test pAKT on these samples. We will expand to additional cohorts and institutions in future studies to address these questions.

3. Although the extension to human breast cancers in Figs. 3G-H and Fig. 7 is important, there are concerns. For example, Fig. 3G indicates that "tissue sections from PARPi-resistant tumors" were utilized in Fig. 3H. However, Fig. 3H stratifies progression-free survival in these PARPi-resistant tumors by high or low levels of pFLT1. If all of the tumors are PARPi-resistant, maybe differences in pFLT1 (FLT1 activation) are not driving resistance but merely controlling the growth rate of the tumors. Can clarification be added to address this point?

Authors: We agree this could be confounding unless we have matched pre- and post-treatment tissues. Therefore, we have removed these data from the manuscript.

Also, while Suppl. Tables 1 and 2 indicate that human breast cancer samples utilized in Figures 3G-H and 7, respectively, had mutations in either BRCA1, BRCA2 or PALB2, the Results section and the figure legends do not appear to indicate which of the high vs low pFLT1 tumors had mutations in each gene. This is particularly important since the only extension of the results in mice for Brca1- and Bard1-deficiency to these other key HDR genes (BRCA1 and PALB2) is in these experiments utilizing human tumors.

Authors: We have now provided this information in Table EV1.

Additionally, the presentation for these human tumors would be strengthened by showing examples (images) of tumors with high vs low pFLT1.

Authors: We have now provided this information in Fig. 7B.

Minor issues:

1. Why, as shown in Figs. 1B&E do Bard1-deficient tumors grow more rapidly than Brca1-deficient tumors in the absence of PARPi?

Authors: We agree with the reviewer that there are possibly inherent differences between the two tumor cell lines in growth patterns, which we have not explored. For instance, *Bard1*-def tumors induce cachexia (our published studies, PMID 32730698) whereas *Brca1*-def tumors do not (mentioned in the Results section). At present, we do not know the underlying mechanisms behind these differences.

2. In many figures, such as in Figs. 2B, 2D, 2F and 3B the scales are different, sometimes dramatically so, when comparing Brca1-deficient vs Bard1-deficient cells/tumors in the same figure part. At a minimum, the authors should acknowledge this and give some justification for presenting the data in this way.

Authors: The scales change for each figure to make it easier for the reader to visualize the fold change between experimental and control groups. The data is normalized to the control group, thus setting the control to 1 in each case. The values for each of the experimental groups are compared relative to their respective control groups. The magnitude of the scale is reflective of the relative expression of the experimental group over control expression. We have clarified this now in the Methods section.

3. While both Brca1-deficient and Bard1-deficient tumors show clear differences between the growth/survival of resistant vs sensitive cells in vivo, they argue the differences are less pronounced in vitro (Suppl. Fig. 1). In actuality, in vitro, it looks as though there are clear and significant differences for Brca1-deficient cells but not Bard1-deficient tumor cells. Rather than indicating (as in the title to Suppl. Fig. 1) that both Brca1- and Bard1-deficient cells show "modest differences in talazoparib sensitivity", it might be clearer to acknowledge that these two genotypes do not appear to behave in the same way in vitro.

Authors: We have updated the title and results section to reflect this point.

4. Either the text of the Results section or the legend to figure 2 should clarify how endothelial cells are being identified for this figure, on the basis of being CD31+, by morphology, or by both.

Authors: We have updated the figure legend to indicate that the endothelial cells are being identified by both CD31 immunostaining and morphology.

5. Suppl. Figure 2 compares Brca1- and Bard1-deficient tumors that are sensitive or resistant to PARPi for immune cell composition. The authors should acknowledge that for certain types of immune cells, such as B cells and macrophages, that there are not consistent differences between sensitive and resistant tumors in the two distinct genotypes.

Authors: We have updated the Results section to address this point.

6. While Suppl. Fig. 3B quantifies the number of FLT1+ cells in a specific area, based on the examples shown in Supp. Fig. 3A, it would appear that the mean intensity of FLT1 is much higher in PARPi resistant tumors from a Brca1-deficient genetic background than for Bard1-deficient tumors. Either more representative examples are needed or there some clarification should be added.

Authors: Since our text appears confusing, we would like to clarify a few points. We have updated this in our Methods section as well.

(a) In Fig 3A-B, EV3A-B and all figures with image analysis, we have quantified the entire tissue sections (and not specific fields) from all mice shown using the Qu Path software.

(b) The Qu Path software used in this study enables the quantification of the signal intensity of a particular immunostaining across the entire section. Using this software, the positive cells of the selected signal intensity are quantified. The final data were generated by dividing the number of positive cells (of the selected staining intensity) by the total area of the tumor section, and the results are normalized to the comparator to present the data as a fold change relative to their respective control (which is set to 1). These images and staining intensities are manually examined by a trained pathologist.

Therefore, the intensity differences between *Brca1*- or *Bard1*- deficient tumors do not impact the quantification, since the intensities of the experimental groups are compared to their own respective controls within each model. This is also described in detail in the Methods section.

7. Neither the text of the Results section nor the legend to Fig. 6 appears to indicate what model of T-cell deficient mice is utilized.

Authors: We have used the athymic nu/nu (nude-*Foxn1*^{nu}) strain, which is now indicated in the Results and Methods section.

Referee #3 (Comments on Novelty/Model System for Author):

Responses to Reviewer 3 (in blue).

The experiments reported here are mainly performed using in vivo models. As also suggested in my report, the

use of organoids derived from BC patients may be an alternative for some experiments. However, I believe that the authors have done a great job and the results are very interesting and with a great potential.

Referee #3 (Remarks for Author):

The authors identify a novel mechanism underlying the resistance to PARPi in breast cancers, that through the activation of the FLT1 and AKT leads to cell survival of BRCA1/2-mutant breast cancer. The experiments are performed mostly in in vivo models, that is remarkable, and strongly support the conclusions drawn in the paper. The text is well-written and straightforward. The topic they face is of great importance. Therefore, overall, I believe that the work presented is suitable for publication in EMM.

Authors: We appreciate these encouraging comments on the novelty and importance of these *in vivo* drug resistance modeling studies.

However, I have some comments/suggestions and requests, more specifically from a mechanistic point of view.

The data clearly demonstrate that AKT is activated, conferring pro-survival signals. I suppose that the authors checked the activation of the S473 residue (since this ab has been listed in the "Material" section).

1. Can the authors indicate the specific residue?

Authors: We thank the reviewer for pointing out this missing information. The pAKT residue is indeed Ser 473, and we have now updated the figures to make this obvious to the reader.

2. What about the activation of the T308?

Authors: We saw a similar activation of Thr308 in resistant tumors by immunostaining analysis (shown in *Reviewer's only Fig. 1* at the end of this document).

3. Does FLT1 directly activate AKT? On the same line, and considering that the PI3K pathway is a well-recognized mechanism of BC resistance, can the authors clarify if there is any link between these two pathways?

Authors: There is prior literature showing FLT1 binding to the p85 subunit of PI3K *in vitro* (PMID: 7657594). Therefore, it is possible that FLT1 activation of AKT is not direct and could act through PI3K. In future studies, we will determine the binding partners of FLT1, explore whether FLT1 directly binds to PI3K or AKT and identify the upstream and downstream signaling mechanisms to AKT. Our present study is focused on demonstrating that PGF-induced FLT1 activation in cancer cells promotes PARP inhibitor resistance in breast cancer models. We will mechanistically address how FLT is activated in future follow-up studies.

Alpelisib is a well-known drug used in combination with PARPi.

Did the authors check if alpelisib+PARPi phenocopy the effects given by axitinib+PARPi? And, more importantly, do AKT inhibitors (e.g. capivasertib) phenocopy the effects given by axitinib+PARPi? This would strongly demonstrate that the effects here described are through the FLT1-AKT pathway.

Authors: We previously informed the editor that the syngeneic mouse strain used for implanting the *Brca1*- and *Bard1*-deficient cells had been unavailable for several months, which caused the delay in our revision experiments. In the end, we were only able to obtain age- and sex-matched mice in limited quantities. Therefore, we prioritized the most essential experiments for this revision. Following the reviewer's recommendations, we performed in-vivo tumor assays with the AKT inhibitor (capivasertib) shown below in *Reviewer's only Figure. 2* at the end of this document.

Consistent with the FLT1 inhibition phenotypes, blockade of AKT in combination with PARP inhibitor results in tumor regression of PARP-inhibitor-resistant tumors in the *Brca1*-deficient model. However, we did not have sufficient mice to repeat these experiments in the *Bard1*-deficient model.

Moreover, are there additional targets of the FLT1 activation that contribute to the effects you describe? Maybe a phospho-proteomic analysis may be helpful.

Authors: We appreciate the suggestion and plan to conduct phospho-proteomic analysis in future studies to identify other targets of FLT1 activation beyond AKT.

An additional point I would like to highlight is on the use of the PARP inhibitor. The PARP inhibitor used by the authors is talazoparib, that can block the activity of both PARP1 and PARP2. Recent literature shows that inhibition of PARP2 caused side effects at hematopoietic level; a novel and highly selective PARP1 inhibitor is now available (AZD530). I would suggest to add the use this inhibitor in key experiments.

Authors: We appreciate this insightful comment. In our studies, since resistance to PARP inhibitors is generated *in vivo* by long term daily treatment of mice, we would need to start from the beginning with the new drug AZD530 by first generating PARP inhibitor in-vivo resistance models. This process along with the subsequent validation of the targets could take anywhere from 1-2 years for completion. Therefore, we feel this particular experiment is out of scope for the present study but will be important to perform in future follow-up translational studies.

To further strength their findings, could the author use the proposed therapy in organoids derived from BC patient resistant to PARPi?

Authors: Since the patient samples from BRCA1/2-mutant breast cancer patients are relatively rare, organoid models are not available. Fresh tumor samples are difficult to obtain, which further limits the possibilities of readily generating organoid models, which we agree would be very useful resources for validation.

Additional points:

Please see Figs 1 and 2 (reviewers only) in the next pages.

Title of the manuscript was updated.

A new co-author was added (Yifan Gu, technician in the Acharyya laboratory) to assist with the revision experiments.

REVIEWERS: Figure 1

Figure 1. Activation of AKT (Thr 308) in PARPi-resistant *Brca1*- and *Bard1*-def breast tumors. pAKT (Thr308) staining was quantified using automated QuPath software to identify positively stained cells in the indicated tumor sections. $n = 4$ for Sen and Res tumors for both *Brca1*-def and *Bard1*-def models. Data are presented as mean values \pm SEM. P values were determined by a two-tailed, unpaired, Mann-Whitney test: * indicates $P = 0.0286$ for both models.

Figure 2. Pharmacological inhibition of AKT with Capiasertib with Talazoparib phenocopies the tumor-inhibitory effect of axitinib+talazoparib treatment. **A.** Mice were injected with the Tal-resistant (“Res”) *Brca1*-def breast tumor cells described in Fig. 1 and then randomized into the following six treatment groups at 2 weeks post tumor-cell injection: 1) vehicle (“Veh”), 2) talazoparib (“Tal”), 3) axitinib (“Axi”), 4) Tal + Axi, 5) Capiasertib (“Cap”), and 6) Tal + Cap. Tumor size was measured weekly to monitor tumor growth, and mice were euthanized for tumor collection at 4 weeks post tumor-cell injection. $n = 6$ Veh-treated tumors, $n = 7$ Tal-treated tumors, $n = 5$ Axi treated tumors, $n = 5$ tumors treated with Tal + Axi, $n = 6$ Cap-treated tumors, and $n = 6$ tumors treated with Tal + Cap. Veh, Tal, Axi and Tal+ Axi data shown from Fig. 4 and EV4. Data are presented as mean values \pm SEM. P values were determined with a one-way ANOVA test. The endpoint tumor volume between the Tal and Tal + Axi groups is **** at 4 weeks, indicating $P < 0.0001$. The endpoint tumor volume between the Tal and Tal + Cap groups is **** at 4 weeks, indicating $P < 0.0001$. **B.** Tumor weights from A were plotted at endpoint. $n = 6$ Veh-treated tumors, $n = 7$ Tal-treated tumors, $n = 5$ Axi-treated tumors, $n = 5$ tumors treated with Tal + Axi, $n = 6$ Cap-treated tumors, and $n = 6$ tumors treated with Tal + Cap. Data are presented as mean values \pm SEM. P values were determined by a two tailed, unpaired, Mann-Whitney test. ** indicated $P = 0.0051$ between the Tal and Tal + Axi groups and ** indicates $P = 0.0012$ between the Tal and Tal + Cap groups.

29th May 2024

Dear Dr. Acharyya,

Thank you for submitting your revised study. Your manuscript was sent back to the initial referees. Referee #2 was unfortunately not available, but referee #3 also evaluated your answers to this referee's concerns. As you will see below, they are satisfied with the revisions, and I will therefore be able to accept your manuscript once the following editorial points will be addressed:

1/ Manuscript text:

- Please remove the coloured font, and only keep in track changes mode any new modification.
- We can accommodate a maximum of 5 keywords, please adjust accordingly.
- Please remove "data not shown" (p.6): as per our guidelines on "Unpublished Data" the journal does not permit citation of "Data not shown". All data referred to in the paper should be displayed in the main or Expanded View figures.
- Methods:
 - o Patient samples: please include a statement that informed consent was obtained from all subjects.
 - o Please add a Graphics section, mentioning the use of Biorender: "(some of the... OR Figure #... OR synopsis) Graphics were created with BioRender.com".
 - Data availability: please remove the current text and indicate "This study includes no data deposited in external repositories." (see also: <https://www.embopress.org/page/journal/17574684/authorguide#availabilityofpublishedmaterial>)
 - The acknowledgement section should be placed after the Methods section, and match the information provided in the submission system (currently HICCC Pilot Awards, Columbia University Irving Scholars Program (and P30CA013696?) are missing in the submission system).
 - A "Disclosure statement and competing interests" section should be included after the acknowledgements (<https://www.embopress.org/competing-interests>).
 - The References should be listed in alphabetical order, with 10 author names before et al.

2/ Figures and Appendix:

- Please address the queries from our data editors in the figure legends:
 1. Although 'n' is provided, please describe the nature of entity for 'n' in the legends of figures 2b, d, f, h, j; 3b; 5c, f; 6b-c, e, g, j; EV 2a-d; EV 3b, d-i, k-l; EV 5d.
 2. Please note that the error bars are not defined in the legends of figures 1c, f; EV 2c.
 3. Please note that the scale bar needs to be defined for figures 2a, c, e, g; 3a; 5c, e, g; 6a, f; 7b; EV 3a, c.
- Appendix: please add page numbers (including in the table of content). Ideally the legends should be placed underneath the corresponding figure and table. Table should be renamed "Appendix Table S1"
- Thank you for providing Source Data. Please check the provided SD values for Figure 3D, Flt1i (gRNA1) + Veh (week 1 and 2 identical).

4/ I introduced minor modifications to your synopsis, let me know if you agree with the following or amend as you see fit:

PARP inhibitor (PARPi) resistance is a major treatment challenge that dramatically shortens patient survival. Using new mouse models of PARPi response and recurrence, we identified FLT1 as a potential biomarker and therapeutic target for reversing PARPi resistance in BRCA deficient-breast cancer.

- New mouse models were developed that recapitulate the PARPi response and recurrence observed in patients.
- A novel PARPi-adaptive resistance mechanism driven by the PGF-FLT1-AKT pathway was identified.
- FLT1 signaling protected the cells from PARPi-induced death by activating AKT pro-survival pathways and by dampening the cytotoxic immune response.
- Blocking FLT1 signaling, either genetically or pharmacologically using axitinib, re-sensitized PARPi-resistant tumors to PARPi treatment in mice.
- High FLT1 activation in tumor cells at pre-treatment significantly correlated with shorter progression-free survival on PARPi in patients with breast cancer.

Thank you for providing a nice synopsis image. Please resize it to 550 px wide x 300-600 px high and make sure that the text remains legible.

5/ As part of the EMBO Publications transparent editorial process initiative (see our Editorial at <http://embomolmed.embopress.org/content/2/9/329>), EMBO Molecular Medicine will publish online a Review Process File (RPF) to accompany accepted manuscripts.

This file will be published in conjunction with your paper and will include the anonymous referee reports, your point-by-point response and all pertinent correspondence relating to the manuscript. Let us know whether you agree with the publication of the RPF and as here, if you want to remove or not any figures from it prior to publication.

I look forward to receiving your revised manuscript.

With kind regards,

Lise Roth

Lise Roth, PhD

Senior Editor

EMBO Molecular Medicine

***** Reviewer's comments *****

Referee #1 (Comments on Novelty/Model System for Author):

The revised manuscript addressed the critiques. The models used are adequate.

Referee #1 (Remarks for Author):

The authors have addressed my previous comments. I have no additional concerns.

Referee #3 (Remarks for Author):

The authors addressed most of the points raised by reviewers. They have performed a meticulous revision of the manuscript, providing strong evidence of the importance of FLT1 activation in promoting resistance to PARPi in BC. I believe the manuscript is suitable for publication in EMM Journal.

June 4, 2024

Dear Dr. Roth,

Thank you for giving us the opportunity to resubmit our manuscript. We have addressed all the points that you mentioned below, and the point-by-point responses are included.

Thank you,
Swarnali

1/ Manuscript text:

- Please remove the coloured font, and only keep in track changes mode any new modification.

Authors: We have made these changes.

- We can accommodate a maximum of 5 keywords, please adjust accordingly.

Authors: We have made these changes and updated the manuscript.

- Please remove "data not shown" (p.6): as per our guidelines on "Unpublished Data" the journal does not permit citation of "Data not shown". All data referred to in the paper should be displayed in the main or Expanded View figures.

Authors: we have removed them.

- Methods:

o Patient samples: please include a statement that informed consent was obtained from all subjects.

Authors: Completed.

o Please add a Graphics section, mentioning the use of Biorender: "(some of the... OR Figure #... OR synopsis) Graphics were created with [BioRender.com](https://biorender.com)".

Authors: Completed.

- Data availability: please remove the current text and indicate "This study includes no data deposited in external repositories." (see

also: <https://www.embopress.org/page/journal/17574684/authorguide#availabilityofpublishedmaterial>)

Authors: Completed.

- The acknowledgement section should be placed after the Methods section, and match the information provided in the submission system (currently HICCC Pilot Awards, Columbia University Irving Scholars Program (and P30CA013696?) are missing in the submission system).

Authors: The relocation of the acknowledgement section has been done. All grants supporting the study and additional ones (supporting my salary during this study) have been added.

- A "Disclosure statement and competing interests" section should be included after the acknowledgements

(<https://www.embopress.org/competing-interests>).

Authors: Completed.

- The References should be listed in alphabetical order, with 10 author names before et al.

Authors: They have been updated using Endnote (EMBO Mol Med Style).

2/ Figures and Appendix:

- Please address the queries from our data editors in the figure legends:

1. Although 'n' is provided, please describe the nature of entity for 'n' in the legends of figures 2b, d, f, h, j; 3b; 5c, f; 6b-c, e, g, j; EV 2a-d; EV 3b, d-i, k-l; EV 5d.

Authors: Updated.

2. Please note that the error bars are not defined in the legends of figures 1c, f; EV 2c.

Authors: Updated.

3. Please note that the scale bar needs to be defined for figures 2a, c, e, g; 3a; 5c, e, g; 6a, f; 7b; EV 3a, c.

Authors: Updated.

- Appendix: please add page numbers (including in the table of content). Ideally the legends should be placed underneath the corresponding figure and table. Table should be renamed "Appendix Table S1"

Authors: Updated.

- Thank you for providing Source Data. Please check the provided SD values for Figure 3D, Flt1i (gRNA1) + Veh (week 1 and 2 identical).

Authors: It is the same for both. We checked the data again.

4/ I introduced minor modifications to your synopsis, let me know if you agree with the following or amend as you see fit:

PARP inhibitor (PARPi) resistance is a major treatment challenge that dramatically shortens patient survival. Using new mouse models of PARPi response and recurrence, we identified FLT1 as a potential biomarker and therapeutic target for reversing PARPi resistance in BRCA deficient-breast cancer.

- New mouse models were developed that recapitulate the PARPi response and recurrence observed in patients.
- A novel PARPi-adaptive resistance mechanism driven by the PGF-FLT1-AKT pathway was identified.
- FLT1 signaling protected the cells from PARPi-induced death by activating AKT pro-survival pathways and by dampening the cytotoxic immune response.
- Blocking FLT1 signaling, either genetically or pharmacologically using axitinib, re-sensitized PARPi-resistant tumors to PARPi treatment in mice.
- High FLT1 activation in tumor cells at pre-treatment significantly correlated with shorter progression-free survival on PARPi in patients with breast cancer.

Authors: We agree to these changes.

Thank you for providing a nice synopsis image. Please resize it to 550 px wide x 300-600 px high and make sure that the text remains legible.

Authors: This was completed.

5/ As part of the EMBO Publications transparent editorial process initiative (see our Editorial at <http://embomolmed.embopress.org/content/2/9/329>), EMBO Molecular Medicine will publish online a Review Process File (RPF) to accompany accepted manuscripts.

This file will be published in conjunction with your paper and will include the anonymous referee reports, your point-by-point response and all pertinent correspondence relating to the manuscript. Let us know whether you agree with the publication of the RPF and as here, if you want to remove or not any figures from it prior to publication.

Authors: We agree to this and there is no restriction on the figures or texts to be published online.

7th Jun 2024

Dear Dr. Acharyya,

Thank you for sending your revised files. I am pleased to inform you that your manuscript is accepted for publication and is now being sent to our publisher to be included in the next available issue of EMBO Molecular Medicine!

With kind regards,

Lise
